# Redirecting antibody responses from egg-adapted epitopes following repeat vaccination with recombinant or cell culture-based versus egg-based influenza vaccines

Feng Liu [1], F. Liaini Gross[1], Sneha Joshi[1], Manjusha Gaglani [2,3,4], Allison L. Naleway[5], Kempapura Murthy[4], Holly C. Groom[5], Meredith G. Wesley[1,6], Laura J. Edwards[6], Lauren Grant[1], Sara S. Kim[1], Suryaprakash Sambhara[1], Shivaprakash Gangappa [1], Terrence Tumpey[1], Mark G. Thompson[1], Alicia M. Fry[1], Brendan Flannery [1], Fatimah S. Dawood[1] & Min Z. Levine [1] ✉

Repeat vaccination with egg-based influenza vaccines could preferentially boost antibodies targeting the egg-adapted epitopes and reduce immunogenicity to circulating viruses. In this randomized trial (Clinicaltrials.gov: NCT03722589), sera pre- and post-vaccination with quadrivalent inactivated egg-based (IIV4), cell culture-based (ccIIV4), and recombinant (RIV4) influenza vaccines were collected from healthcare personnel (18-64 years) in 2018–19 (*N* = 723) and 2019–20 (*N* = 684) influenza seasons. We performed an exploratory analysis. Vaccine egg-adapted changes had the most impact on A(H3N2) immunogenicity. In year 1, RIV4 induced higher neutralizing and total HA head binding antibodies to cell- A(H3N2) virus than ccIIV4 and IIV4. In year 2, among the 7 repeat vaccination arms (IIV4-IIV4, IIV4-ccIIV4, IIV4-RIV4, RIV4-ccIIV4, RIV4-RIV4, ccIIV4-ccIIV4 and ccIIV4-RIV4), repeat vaccination with either RIV4 or ccIIV4 further improved antibody responses to circulating viruses with decreased neutralizing antibody egg/cell ratio. RIV4 also had higher post-vaccination A(H1N1)pdm09 and A(H3N2) HA stalk antibodies in year 1, but there was no significant difference in HA stalk antibody fold rise among vaccine groups in either year 1 or year 2. Multiple seasons of non-egg-based vaccination may be needed to redirect antibody responses from immune memory to egg-adapted epitopes and re-focus the immune responses towards epitopes on the circulating viruses to improve vaccine effectiveness.

Influenza viruses are a common cause of respiratory infections in humans, annual influenza vaccination has been recommended in the United States (US) for many decades. Conventionally, influenza vaccine seed viruses are propagated in embryonated chicken eggs to produce inactivated influenza vaccines (IIV). However, this manufacturing process can lead to egg-adapted mutations in the hemagglutinin (HA) protein in IIV, that could result in altered antigenicity and reduced vaccine effectiveness (VE) against the circulating influenza strains[1-6]. Therefore, vaccine manufacturing platforms that do not rely on eggs have been established in the past decade including

a cell culture-based influenza vaccine (Flucelvax Quadrivalent™ by Seqirus, Inc., ccIIV4) and a recombinant influenza vaccine (Flublok Quadrivalent® by Sanofi Pasteur, RIV4). RIV4 contains 3 times the antigen dose at 45 µg HA/dose/strain compared to the standard dose IIV4 and ccIIV4 (15 µg HA/dose/strain). Both ccIIV4 and RIV4 are currently licensed in the US[7]. Clinical trials evaluating ccIIV and RIV have shown comparable to advantageous immunogenicity compared to standard-dose egg-based IIVs in different age groups[8–13]. Observational studies and clinical trials reported that ccIIV and RIV demonstrated modest to significant improvement of VE over egg-based IIV in different age groups and influenza seasons[7,14–16].

Repeat annual influenza vaccination is another factor that may affect the antibody responses elicited by influenza vaccines. Studies indicated that frequent prior vaccination could blunt the antibody responses to influenza vaccination over time[11,13,17–21] which can lead to reduced vaccine effectiveness[22–26]. One study reported that repeat vaccination reduced antibody affinity maturation to the HA antigens of all vaccine components from egg-based, cell-culture based, and recombinant influenza vaccines[20]. Thus far, there are few direct comparisons of immunogenicity of the three vaccine platforms in populations with a history of frequent influenza vaccinations. We recently reported the comparison of immunogenicity of ccIIV and RIV to egg-based standard dose IIVs among US healthcare personnel (HCP) aged 18−64 years from a randomized, open-label trial during the 2018−2019 season[27] and the 2019−2020 season following repeat vaccination in 2 consecutive years[28,29]. Those comparisons were focused on the neutralizing antibody responses to cell-propagated vaccine viruses only. RIV4 elicited higher neutralizing antibody responses than IIV4 against cell-propagated vaccine viruses[27]. Repeat vaccination with ccIIV4 or RIV4 in 2 seasons also resulted in significantly higher post-vaccination antibody responses to cell-propagated vaccine viruses[29]. Most HCP participants had received egg-based influenza vaccines frequently during the preceding five seasons.

We previously reported that antibodies targeting the egg-adapted epitopes could be preferentially boosted from repeat vaccination with egg-based vaccines[3]. The potential of non-egg-based vaccines to mitigate the impact of prior egg-based vaccination remains unclear. In this study, we analyzed neutralizing antibody responses to both egg- and cell-propagated vaccine viruses, and total binding antibodies to both HA head and HA stalk among HCP following vaccination with RIV, ccIIV, and standard egg-based IIVs in two influenza seasons to investigate whether repeat vaccination with non-egg-based vaccines can overcome the effect of prior repeat vaccination with egg-based vaccines.

## Results

### Influenza vaccines and vaccine viruses used during the 2018−2019 and 2019−2020 study seasons

This study was a randomized, open-label trial conducted in the United States during the Northern Hemisphere 2018−2019 (Year 1, $n = 723$) and 2019−2020 (Year 2, $n = 684$) influenza seasons (Fig. 1). In year 1, HCP were randomized to receive one of four influenza vaccines: ccIIV4 (Flucelvax), RIV4 (Flublok), Fluzone IIV4 or Fluarix IIV4. Year 1 vaccines contain 2018−2019 Northern Hemisphere formulation including an A/Michigan/45/2015 (H1N1)pdm09-like virus; an A/Singapore/INFIMH-16-0019/2016 (H3N2)-like virus; a B/Colorado/06/2017-like virus (B/Victoria/2/87 lineage); and a B/Phuket/3073/2013-like virus (B/Yamagata/16/88 lineage) (Table 1). ccIIV4 in the 2018−2019 season contained cell-culture derived seed strains for the influenza A(H3N2) and B vaccine strains but egg-based seed strain for A(H1N1)pdm09. In year 2 (2019−2020 season), HCP were re-randomized into 7 repeat vaccination arms: IIV4-IIV4, IIV4-ccIIV4, IIV4-RIV4, RIV4-ccIIV4, RIV4-RIV4, ccIIV4-ccIIV4 and ccIIV4-RIV4 (Fig. 1). The A(H1N1)pdm09 and A(H3N2) components were updated to a A/Brisbane/02/2018 (H1N1)pdm09-like virus and an A/Kansas/14/2017 (H3N2)-like virus respectively while the B/Victoria and B/Yamagata components remained the same (Table 1).

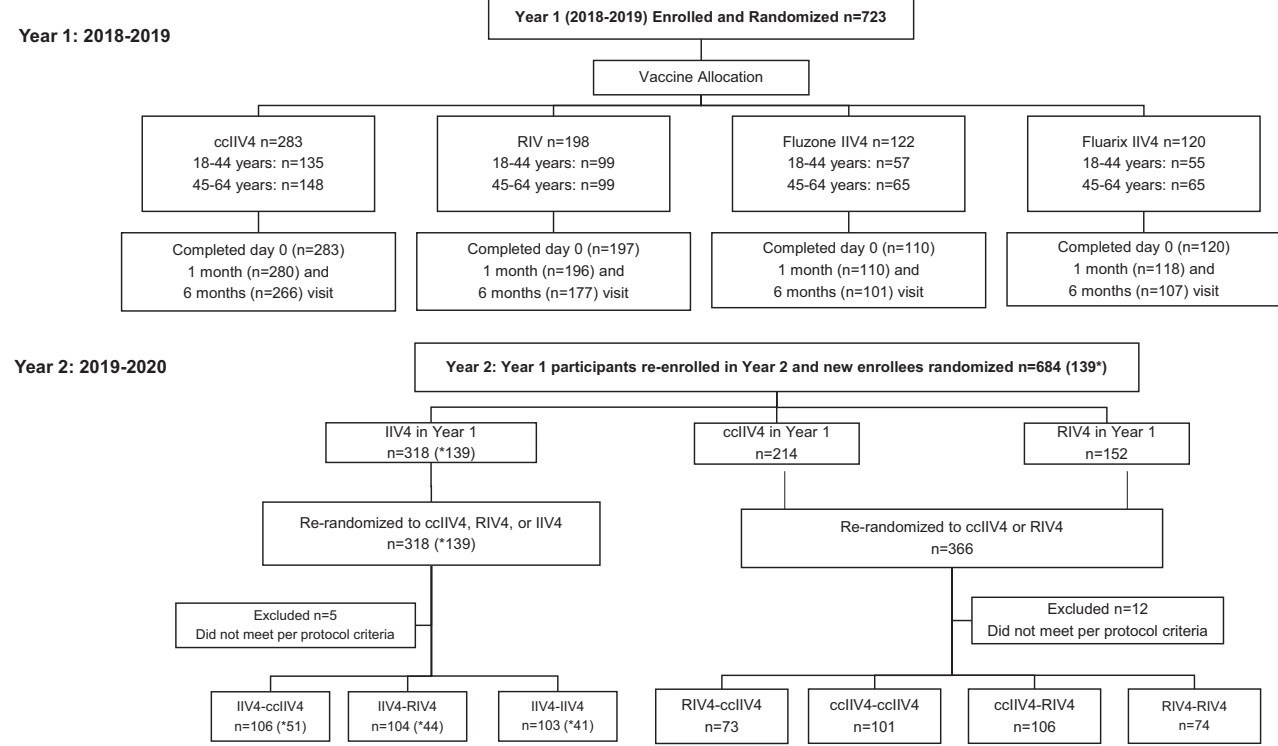

**Fig. 1 | CONSORT diagrams for screening and Enrollment of participants in Year 1 (2018−2019) and Year 2 (2019−2020).** Abbreviations: IIV4: Inactivated influenza vaccine (Fluzone® or Fluarix® Quadrivalent); ccIIV4: Cell-culture based IIV (Flucelvax® Quadrivalent); RIV4: recombinant-hemagglutinin influenza vaccine (Flublok® Quadrivalent). *Numbers in parenthesis are new enrollees for Year 2.

**Table 1 | Vaccine egg-adapted mutations in the HA head domain of egg-grown vaccine viruses and CVVs in year 1 and year 2 of the study**

| Viruses | Amino acids differences on the HA head domain | | | | | | | | | | | | | | | |
|---|---|---|---|---|---|---|---|---|---|---|---|---|---|---|---|---|
| **A(H1N1)pdm09** | 45 | 74 | 120 | 164 | 183 | 209 | 223 | 225 | 282 | 295 | 298 | | | | | |
| Year 1 **A/Michigan/45/2015 cell** [a] | R | S | T | S | S | K | Q | G | P | I | I | | | | | |
| A/Michigan/45/2015 egg | R | S | T | S | S | K | *R* | G | P | I | I | | | | | |
| A/Michigan/45/2015 X-275 egg CVV (EPI830246) [b] | R | S | T | S | S | *M* | *R* | G | P | I | I | | | | | |
| A/Singapore/GP1908/2015 IVR-180 egg CVV (EPI848715) [c,d] | R | S | A | S | S | K | Q | *A* | P | I | I | | | | | |
| Year 2 **A/Idaho/07/2018 cell** [a,d] | R | R | T | T | P | K | Q | G | P | V | I | | | | | |
| A/Brisbane/02/2018 egg [b] | G | R | T | T | P | K | *R* | G | A | V | V | | | | | |
| **A(H3N2)** | 91 | 121 | 128 | 138 | 142 | 144 | 158 | 159 | 160 | 171 | 190 | 193 | 194 | 225 | 246 | 311 |
| Year 1 **A/Singapore/INFIMH-16-0019/2016 cell** [a] | S | K | T | A | G | S | N(CHO) | Y | T | K | D | F | L | D | N | H |
| A/Singapore/INFIMH-16-0019/2016 egg | S | K | T | A | G | S | *N(CHO-)* | Y | *K* | K | D | F | *P* | *D/G*[¶] | N | H |
| A/Singapore/INFIMH-16-0019/2016 IVR-186 egg CVV (EPI1082512) [b] | S | K | T | A | G | S | *N(CHO-)* | Y | *K* | K | D | F | *P* | *G* | N | H |
| A/Singapore/INFIMH-16-0019/2016 NIB-104 egg CVV (EPI1151864) [c] | S | K | T | A | G | S | *N(CHO-)* | Y | *K* | K | D | F | *P* | *G* | N | H |
| A/North Carolina/04/2016 cell (EPI701956) [d] | S | N | T | A | R | S | N(CHO) | Y | T | K | D | F | L | D | N | H |
| Year 2 **A/Kansas/14/2017 cell** [a,d] | N | N | A | S | G | K | N(CHO-) | S | K | N | D | S | L | D | N | Q |
| A/Kansas/14/2017 egg [b] | N | N | A | S | G | K | N(CHO-) | S | K | N | *N* | S | L | D | *T* | Q |
| **B/Victoria lineage** | 129 | 197 | 199 | | | | | | | | | | | | | |
| Year 1 & 2 **B/Colorado/06/2017 cell** | G | N | T | | | | | | | | | | | | | |
| B/Colorado/06/2017 egg | G | *T(CHO-)* | T | | | | | | | | | | | | | |
| B/Maryland/15/2016 BX-69A egg CVV (EPI1061863) [b,c] | D | *N(CHO-)* | *I* | | | | | | | | | | | | | |
| B/Maryland/15/2016 cell (CY220998) [a] | D | N | T | | | | | | | | | | | | | |
| B/Iowa/06/2017 cell (EPI1619422) [d] | G | N | T | | | | | | | | | | | | | |
| **B/Yamagada lineage** | 154 | 173 | 197 | 252 | | | | | | | | | | | | |
| Year 1 & 2 **B/Phuket/3073/2013 cell** [a] | A | L | N | M | | | | | | | | | | | | |
| B/Phuket/3073/2013 egg [b,c] | A | L | *D(CHO-)* | M | | | | | | | | | | | | |
| B/Singapore/INFTT-16-0610/2016 cell (EPI1843276) [d] | S | Q | N | V | | | | | | | | | | | | |

[a]HA proteins of these vaccine viruses were included in the RIV4 vaccine.
[b]CVV: candidate vaccine virus used in the Fluzone IIV4.
[c]CVV used in the Fluarix IIV4 vaccine.
[d]CVV used in the ccIIV4 vaccine. For A(H1N1)pdm09 virus, the egg CVV was passaged in qualified MDCK cells for manufacturing ccIIV4 used in Year 1.
[¶] Mix of D and G.
CHO: Glycosylation motif. CHO-: Loss of glycosylation motif.
Cell grown vaccine viruses used as reference are bolded. HA substitution due to egg-adaptions are italic and bold.
Accession numbers were noted to those reference sequences downloaded from GISAID database.

## Egg-adapted amino acid substitutions in egg-based vaccines in both study years

HA sequences from components of IIV4, ccIV4 and RIV4 vaccines used in both study years were first analyzed (Table 1). Within the same season, the study vaccines contained different candidate vaccine viruses (CVVs) that are antigenically-like the prototype vaccine viruses recommended by the World Health Organization (WHO)[30]. Compared to the cell-grown vaccine prototype viruses, the vaccine antigens from all 4 subtypes in the egg-based vaccines had HA substitutions due to egg-adaptation. A(H1N1)pdm09 egg CVV formulated in Fluzone IIV4 had Q223R substitution in both years, while the egg CVV used for Fluarix IIV4 and ccIIV4 in year 1 had G225A substitution instead (Table 1). For A(H3N2), year 1 egg-based vaccine had T160K (resulting in a loss of glycosylation motif at HA 158), L194P, and D225G egg-adapted changes, while year 2 egg-based vaccine had D190N and N246T substitutions. Influenza B vaccine viruses remained the same in both years. Egg-based B/Victoria vaccine had T199I substitution, which can cause a loss of glycosylation motif at HA position 197; whereas egg-based B/Yamagata vaccine had

a N197D substitution that can also cause a loss of glycosylation motif at HA position 197.

## Antibody responses at pre-vaccination, 1-month and 6-months post-vaccination in Year 1

We first measured neutralizing antibody responses by microneutralization assays (MN) for A(H3N2) and hemagglutination inhibition assays (HI) for A(H1N1)dpm09 and Bs. In year 1 at baseline, pre-vaccination MN/HI geometric mean titers (GMTs) were similar ($p > 0.05$) among the 4 vaccine groups (Fig. 2). Significant differences in neutralizing antibody responses were detected at 1-month post-vaccination. For A(H3N2), RIV4 vaccination induced significantly higher ($p < 0.05$) neutralizing antibody responses to the cell-propagated A(H3N2) virus than ccIIV4 and IIV4s (Fluarix and Fluzone) in both age groups; RIV4 group also mounted significantly higher ($p < 0.05$) MN antibody responses to the egg-propagated A(H3N2) virus than ccIIV4 and Fluarix-IIV4. For A(H1N1)pdm09, RIV4 induced significantly higher ($p < 0.05$) post-vaccination HI antibody responses to both egg- (GMT 122) and cell- (GMT 137)

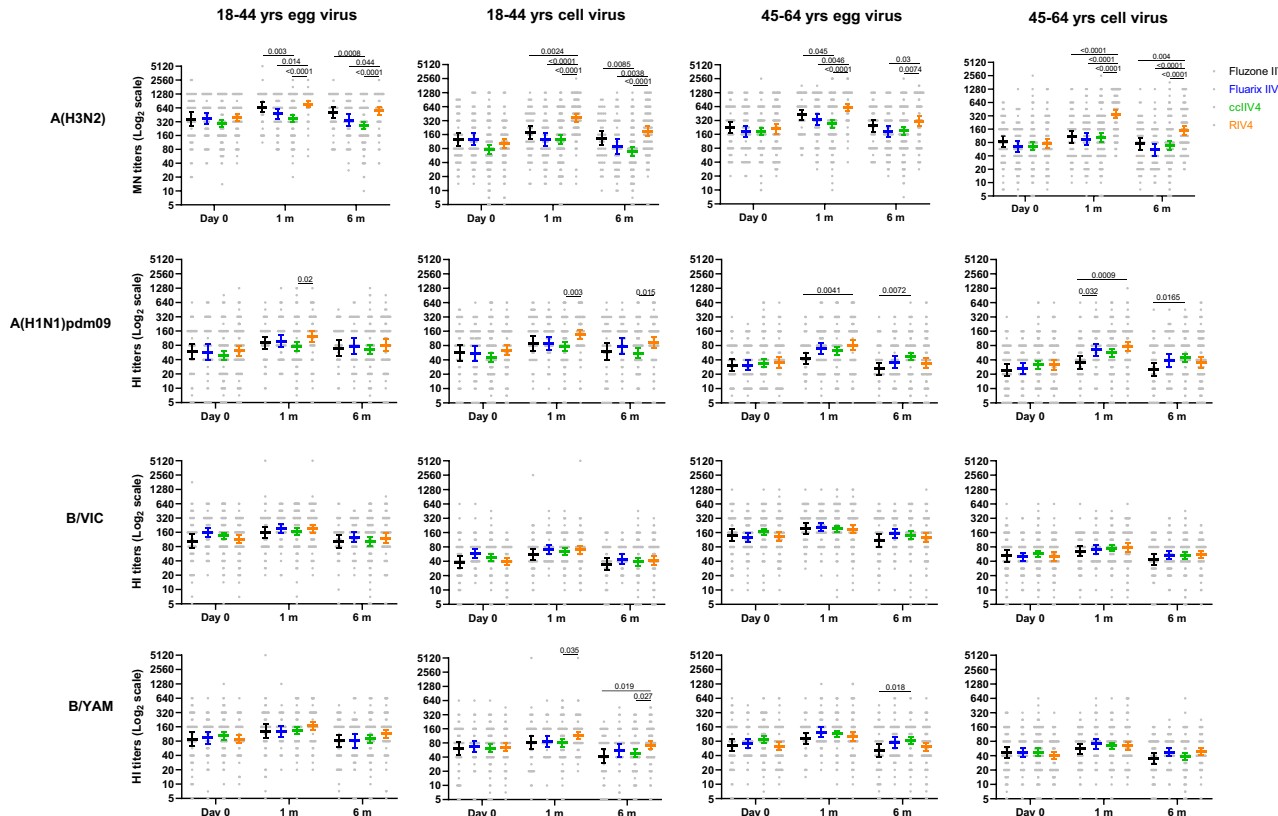

**Fig. 2 | MN or HI Antibody responses to egg- and cell-propagated vaccine viruses at pre-vaccination (Day 0), 1-month, and 6-months post-vaccination in Year 1.** Antibody titers from each of the four vaccine groups were presented as geometric mean titers (GMTs) with 95% confidence interval (CI) respectively for 18−44 years and 45−64 years age groups. Gray dots represented individual titers. Fluzone IIV4 18−44 years: Day 0 ($n = 52$), 1 m ($n = 52$), 6 m ($n = 47$); Fluzone IIV4 45-64 years: Day 0 ($n = 58$), 1 m ($n = 58$), 6 m ($n = 54$); Fluarix IIV4 18−44 years: Day 0 ($n = 55$), 1 m ($n = 55$), 6 m ($n = 48$); Fluarix IIV4 45−64 years: Day 0 ($n = 65$), 1 m ($n = 63$), 6 m ($n = 59$); ccIIV4 18−44 years: Day 0 ($n = 135$), 1 m ($n = 133$), 6 m ($n = 124$);

ccIIV4 45−64 years: Day 0 ($n = 148$), 1 m ($n = 147$), 6 m ($n = 142$); RIV4 18−44 years: Day 0 ($n = 99$), 1 m ($n = 98$), 6 m ($n = 85$); RIV4 45−64 years: Day 0 ($n = 98$), 1 m ($n = 98$), 6 m ($n = 92$); Microneutralization titers (MN) were measured for A(H3N2) virus, while hemagglutination inhibition (HI) titers were measured for A(H1N1) pdm09, B/Victoria (B/VIC), and B/Yamagata (B/YAM) viruses. One-way ANOVA corrected for multiple comparisons (Tukey's test) was used to compare the GMTs of each time point among the 4 vaccine groups. Statistically significant differences between groups are indicated by $p$ values on the horizontal bars.

propagated A(H1N1)pdm09 viruses than ccIIV4 (egg virus GMT 76; cell virus GMT 77) in the 18-44 years age group, and significantly higher ($p < 0.05$) post-vaccination HI antibodies to egg- (GMT 82) and cell- (GMT 76) propagated A(H1N1)pdm09 viruses than Fluzone-IIV4 (egg virus GMT 43; cell virus GMT 36) in the 45-64 years age group. In contrast, post-vaccination HI antibodies to B/Victoria and B/Yamagata viruses were mostly similar ($p > 0.05$) among the 4 vaccines in both age groups, except that RIV4 induced significantly higher ($p < 0.05$) HI antibody responses (GMT: 117) to cell-propagated B/Yamagata vaccine virus than ccIIV4 (GMT: 80) in the 18−44 years age group (Fig. 2). RIV4 also had higher seropositivity rate (proportions of titers ≥40) and seroconversion rate (SCR) than egg-based IIV4 vaccines against the cell-grown A(H3N2) virus (Supplementary Tables S1 and S2)[27].

When comparing post-vaccination antibody responses by fold rise, RIV4 induced overall higher geometric mean fold rise of neutralizing antibody titers to the vaccine viruses (Fig. 3A, B). For A(H3N2), RIV4 induced higher fold rise of neutralizing antibodies to both cell- and egg-propagated A(H3N2) viruses compared to other vaccines in both age groups (Fig. 3A, B). MN fold rise to A(H3N2) cell vaccine virus was significantly higher ($p < 0.0001$) in RIV4 groups (3.5 in 18−44 years, 4.6 in 45−64 years) than those in IIV4 and ccIIV4 groups (range: 1.0−1.6). Neutralizing antibody responses waned at 6-months post-vaccination in year 1. GMTs to each virus at 6-months reduced by 1−2.5 folds comparing with the GMTs at 1-month post-vaccination (Fig. 3C, D). At 6-months post-vaccination, although fold reduction in

neutralizing antibodies (6 month/1 month) was greater in RIV4 recipients to cell-A(H3N2) in both age groups, and to cell-A(H1N1)pdm09 virus in the 45−64 years group (Fig. 3C, D), the titers of neutralizing antibodies to cell-grown viruses in RIV4 recipients still remained at similar or higher levels compared to other vaccine groups (Fig. 2).

To further elucidate the quality of the antibody responses, we then analyzed the total binding antibodies by the enzyme linked immunosorbent assay (ELISA) to both HA head of the 4 antigen subtypes and HA stalk of the A(H3N2) and A(H1N1)pdm09 subtypes in the quadrivalent vaccines. Total binding antibodies include both neutralizing and non-neutralizing antibody responses. For A(H3N2), consistent with MN responses, RIV4 induced significantly higher ($p < 0.01$) total binding antibodies (Fig. 4A) and significantly higher ($p < 0.001$) fold rise (Fig. S1A) to the cell-A(H3N2) HA head at 1-month post-vaccination compared to IIV4 and ccIIV4. For A(H1N1)pdm09, all 4 vaccines induced similar fold rise (1 month/day 0) in the total binding antibodies to egg- and cell- A(H1N1)pdm09 HA (Fig. S1A), although RIV4 group had the highest total binding antibodies to the cell-A(H1N1) pdm09 HA at 1-month post-vaccination, it is likely due to the higher levels of pre-vaccination total binding antibodies (Fig. 4A). For influenza Bs, RIV4 induced significantly higher ($p < 0.01$) fold-rise against B/ VIC and B/YAM cell virus HA head than Fluarix IIV4 but not Fluzone IIV4 (Fig. S1A). At 6-months post-vaccination, total binding antibodies to the HA head waned but the levels of total binding antibodies to the cell-grown virus HA head of all 4 subtypes in the RIV4 group remained the highest among the 4 vaccine groups (Fig. 4A).

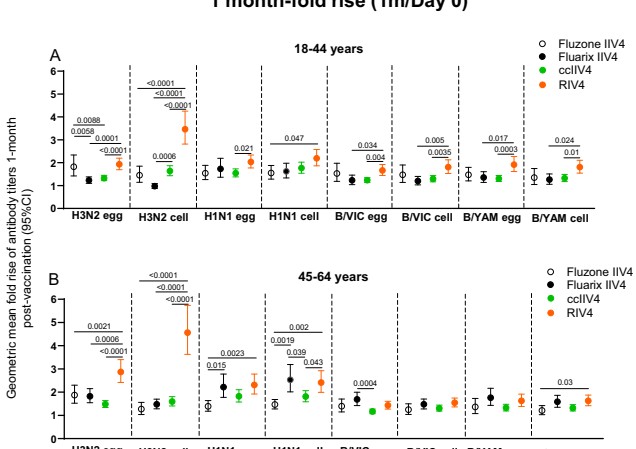

**Fig. 3 | Fold rise of MN or HI antibody titers at 1-month post-vaccination and waning of antibody titers at 6-months post-vaccination in Year 1.** Geometric mean (GM) fold rise of antibody titers (1 m/Day 0) from each of the four vaccine groups were calculated for egg- and cell-propagated vaccine viruses, and for 18–44 years (Fluzone IIV4 (n = 52), Fluarix IIV4 (n = 55), ccIIV4 (n = 133), RIV4 (n = 98)) and 45–64 years (Fluzone IIV4 (n = 58), Fluarix IIV4 (n = 63), ccIIV4 (n = 147), RIV4 (n = 98)) age groups respectively (**A**, **B**). Antibody waning was expressed as the GM fold-reduction of titers from 1-month to 6-months (1 m/6 m) post-vaccination for 18−44 years (Fluzone IIV4 (n = 47), Fluarix IIV4 (n = 48), ccIIV4 (n = 124), RIV4 (n = 85)) and 45−64 years (Fluzone IIV4 (n = 54), Fluarix IIV4 (n = 59), ccIIV4 (n = 142), RIV4 (n = 92)) respectively (**C**, **D**). MN titers were measured for A(H3N2) virus, while HI titers were measured for A(H1N1)pdm09, B/VIC, and B/YAM viruses. One-way ANOVA corrected for multiple comparisons (Tukey's test) was used to compare the GM fold-rise or fold-reduction of each time point among the 4 vaccine groups. Statistically significant differences between groups are indicated by p values on the horizontal bars.

Pre-existing A(H3N2) and A(H1N1)pdm09 HA stalk antibodies were detected in participants at varies levels. Comparing HA stalk antibody titers at 1-month post-vaccination versus day 0, vaccination significantly boosted A(H1N1)pdm09 HA stalk antibodies in all 4 vaccine groups (p < 0.01), and significantly boosted A(H3N2) HA stalk antibodies in 3 of the 4 vaccine groups (p < 0.001, Fluzone IIV4, ccIIV4, and RIV4) (Fig. 4C). The RIV4 group had the highest A(H1N1)pdm09 HA stalk and A(H3N2) HA stalk antibodies at 1-month post-vaccination, however the fold rises of HA stalk antibodies were low (<1.8) and similar among all 4 vaccine groups (Fig. 4D). At 6-months post-vaccination, waning of HA stalk antibodies were more notable in ccIIV4 and RIV4 groups compared to IIV4 (Fig. S2).

Active surveillance was conducted during the influenza season in year 1, a total of 26 breakthrough infection cases were identified with an attack rate of 3.6% (26/723) (Supplementary Table S3).

## Antibody responses to egg-adapted epitopes measured by antibody GMT egg/cell titer ratio in year 1

To assess the levels of antibody responses targeting the egg-adapted epitopes, we used the antibody "GMT egg/cell titer ratio" as a proxy to quantify the extent of difference in antibody responses to egg- versus cell-viruses in each vaccine group (Fig. 5A, B). The post-vaccination neutralizing antibody GMT egg/cell titer ratio was the most pronounced for A(H3N2), with elevated MN GMT egg/cell titer ratios in all vaccine groups (Fig. 5A, B), suggesting vaccination induced significant levels of neutralizing antibody responses targeting A(H3N2) egg-adapted epitopes related to 158 (CHO-), 194 P, and 225 G (Table 1). The post-vaccination A(H3N2) neutralizing antibody GMT egg/cell ratio from the 18−44 years group who received Fluzone IIV4 (3.7), Fluarix IIV4 (3.8), and ccIIV4 (3.0) were significantly higher (p < 0.05) than those in the same age groups who received RIV4 (2.1) (Fig. 5A). Similar trend was also observed in the 45−64 years group, the post-vaccination A(H3N2) neutralizing antibody GMT egg/cell titer ratios in those who received Fluzone IIV4 (4.0), Fluarix IIV4 (3.5), and ccIIV4 (2.6) were significantly higher (p < 0.01) than those who received RIV4 (1.7) (Fig. 5B).

In contrast, the impact of egg-adaption is less profound in HI antibody responses to A(H1N1)pdm09 and both B lineage viruses. For A(H1N1)pdm09 and B/Yamagata, the post-vaccination HI antibody GMT egg/cell titer ratios were all <2 and similar (p > 0.05) among the vaccine groups (Fig. 5A, B). For B/Victoria, the post-vaccination HI GMT egg/cell titer ratios were slightly elevated (range: 2−3) with the ratio in Fluzone IIV4 recipients significantly higher (p < 0.05) than those in RIV4 recipients from the 45−64 years group (Fig. 5B).

Next, to assess whether vaccination with non-egg-based vaccines can overcome the impact of egg-adaptation and reduce the GMT egg/cell ratio in the vaccine responses, we calculated the "proportion of the participants with egg/cell ratio ≥4" in MN/HI titers at pre- and 1-month post-vaccination (Fig. 5C, D). At pre-vaccination, there was no significant difference in the proportion of participants with GMT egg/cell ratio ≥4 among the 4 vaccine groups. At baseline, around 40% of participants had GMT egg/cell ratio ≥4 to A(H3N2) and B/Victoria viruses, suggesting a high prevalence of pre-existing neutralizing antibodies targeting HA egg-adapted epitopes in this highly vaccinated HCP population. After vaccination, in both age groups, the two egg-based vaccines generally reinforced the neutralizing antibody responses to egg-adapted epitopes as evidenced by an increased trend in the proportion of participants with GMT egg/cell titer ratio ≥4. In contrast, among RIV4 and ccIIV4 recipients, the proportion of participants with GMT egg/cell titer ratio ≥4 was overall lower at 1-month post-vaccination compared to pre-vaccination (Fig. 5C, D), demonstrating the potential of non-egg-based vaccines to reduce the neutralizing antibody response difference to egg versus cell viruses. Most notably, only in the RIV4 groups to A(H3N2) virus, the proportions of GMT egg/cell titer ratio ≥4-fold were significantly (p < 0.01) decreased in both age groups (from 58% down to 26% in 18−44 years, and from 40% down to 21% in 45−64 years), suggesting RIV4 vaccination can potentially mitigate the pre-existing antibody memory to egg-adapted epitopes by improving the magnitude of neutralizing antibody responses to cell-propagated viruses (Fig. 5C, D).

We then also analyzed the HA binding antibody "GMT egg/cell titer ratio" measured by ELISA. Similar trend was observed as with MN/HI antibodies that GMT egg/cell ratios were higher (p < 0.05) in the egg-based IIV4 groups than those in the ccIIV4 and RIV4 groups for A(H3N2) HA head and B/VIC HA head binding antibodies (Fig. 4B). However, the GMT egg/cell HA binding antibody ratios (Fig. 4B) were around 1 for all

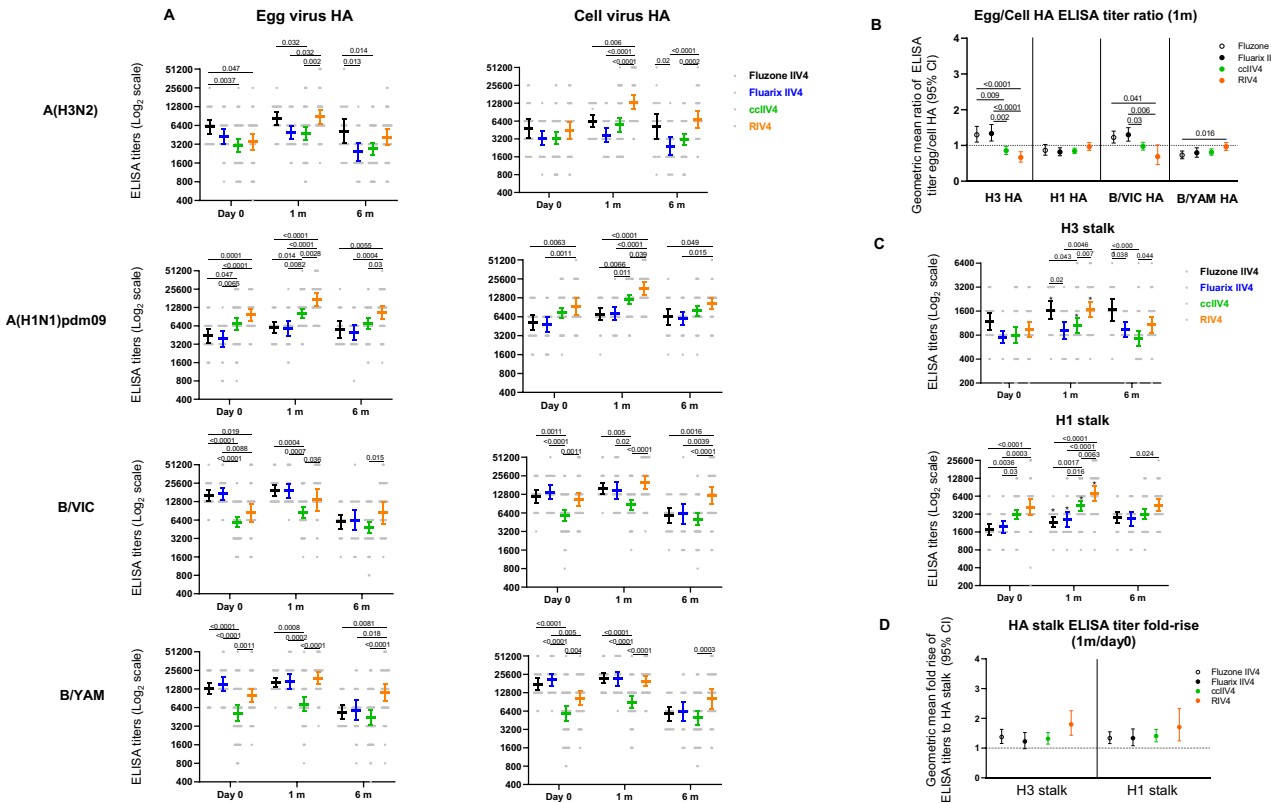

**Fig. 4 | Total HA binding antibody and HA stalk antibody responses measured by ELISA at pre-vaccination (Day 0), 1-month, and 6-months post-vaccination in Year 1.** Antibody titers from each of the four vaccine groups were presented as geometric mean titers (GMTs) with 95% confidence interval (CI) for combined 18−44 years and 45−64 years age groups. Gray dots represented individual titers. Fluzone IIV4: Day 0 (*n* = 24), 1 m (*n* = 24), 6 m (*n* = 21); Fluarix IIV4: Day 0 (*n* = 24), 1 m (*n* = 24), 6 m (*n* = 23); ccIIV4: Day 0 (*n* = 56), 1 m (*n* = 54), 6 m (*n* = 51); RIV4: Day 0 (*n* = 40), 1 m (*n* = 40), 6 m (*n* = 38). **A**: HA (A(H1N1)pdm09) or HA head (A(H3N2), B/VIC, B/YAM) binding antibodies. **B**: Ratios of egg/cell HA or HA head binding antibody titers with 95% CI. **C**: H3 stalk and H1 stalk antibody responses at day 0, 1-month and 6-months post vaccination. **D**: Fold rise of H3 stalk and H1 stalk antibody titers from pre to 1-month post-vaccination with 95% CI. One-way ANOVA

corrected for multiple comparisons (Tukey's test) was used to compare the GMTs of each time point among the 4 vaccine groups (**A**, **C**). One-way ANOVA nonparametric Kruskal-Wallis test was used to compare the fold changes among the 4 vaccine groups (**B**, **D**). Statistically significant differences between groups are indicated by *p* values on the horizontal bars.*: In (**C**), paired *t* test (two-tailed) was used for comparing the pre- and 1 m post-vaccination stalk antibody titers within the same vaccine group: significantly higher HA stalk antibody titers were detected in 1 m post-vaccination than pre-vaccination for H3 stalk binding antibody in Fluzone IIV4 (*p* = 0.0009), ccIIV4 (*p* = 0.0004), RIV4 (*p* < 0.0001); and for H1 stalk binding antibody in Fluzone IIV4 (*p* = 0.0005), Fluarix IIV4 (*p* = 0.0092), ccIIV4 (*p* < 0.0001), RIV4 (*p* = 0.0015).

---

vaccine groups and subtypes, even for A(H3N2), suggesting that unlike the neutralizing antibody responses, most total binding antibody responses do not target the egg-adapted epitopes on HA.

### Redirecting neutralizing antibody responses from egg-adapted epitopes following repeat vaccination with non-egg vs egg-based vaccines in year 2

In year 2, participants who received ccIIV4, RIV4, and Fluzone IIV4 or Fluarix IIV4 in year 1 were re-randomized into seven repeat-vaccination arms: ccIIV4-ccIIV4, ccIIV4-RIV4, RIV4-ccIIV4, RIV4-RIV4, IIV4-ccIIV4, IIV4-RIV4, IIV4-IIV4 (Fluzone) (Fig. 1).

We compared the post-vaccination HI GMTs to the cell-vaccine viruses between IIV4-IIV4 (Fluzone) and each of the 6 repeat vaccination arms with non-egg-based vaccines (Fig. 6). Participants in younger age group (18−44 years) in the ccIIV4-RIV4, RIV4-ccIIV4, RIV4-RIV4, and IIV4-RIV4 arms, and the older age group (45−64 years) in ccIIV4-RIV4 and RIV4-RIV4 arms, all mounted significantly higher (*p* < 0.05) neutralizing antibody titers to cell-A(H3N2) virus than those in the IIV4-IIV4 (Fluzone) arm. For HI antibody response to A(H1N1)pdm09 cell virus, RIV4-RIV4 and IIV4-RIV4 arms in 18−44 years, ccIIV4, RIV4-ccIIV4, RIV4-RIV4 arms in 45−64 years all had significantly higher (*p* < 0.05) HI antibody responses than those in the IIV4-IIV4 (Fluzone) arm. Vaccinees in the 18−44 years group in the ccIIV4-RIV4, RIV4-RIV4, and IIV4-

RIV4 arms had significantly higher (*p* < 0.05) HI antibody responses to both B/VIC and B/YAM cell viruses versus those in the IIV4-IIV4 (Fluzone) arm. Participants in the older age group (45−64 years) who received non-egg-based vaccine in year 2, most had significantly higher (*p* < 0.05) HI GMTs to both B/VIC and B/YAM cell virus compared to those in the IIV4-IIV4 (Fluzone) arm, except ccIIV4-ccIIV4 and IIV4-ccIIV4 arms for B/YAM cell virus (Fig. 6). It is worth noting that those who received IIV4 in Year 1 then RIV4 in Year 2 demonstrated improved antibody responses to cell-grown vaccine viruses compared to those who received IIV4-IIV4 (Fluzone). The seropositivity rates (Supplementary Table S4) and seroconversion rates (Table S5) to both egg- and cell- viruses in year 2 are summarized in the supplementary materials[28,29]. Overall, in year 2 participants who received one or two non-egg-based vaccines mounted higher post-vaccination HI antibody responses to cell-grown vaccine viruses than those who received repeated egg-based IIV4 vaccination in both years (IIV4-IIV4).

At 1-month post-vaccination during Year 1, 21−40% of HCP who received only one season of non-egg-based vaccine including RIV4 still had A(H3N2) GMT egg/cell titer ratio ≥4 (Fig. 5C, D). We therefore investigated whether repeat vaccination with non-egg-based vaccines in year 2 could further reduce the HI GMT egg/cell titer ratio (Fig. 7). Among recipients of IIV4-IIV4 (Fluzone), the proportion of participants with HI GMT egg/cell titer ratio ≥4 generally increased, especially for

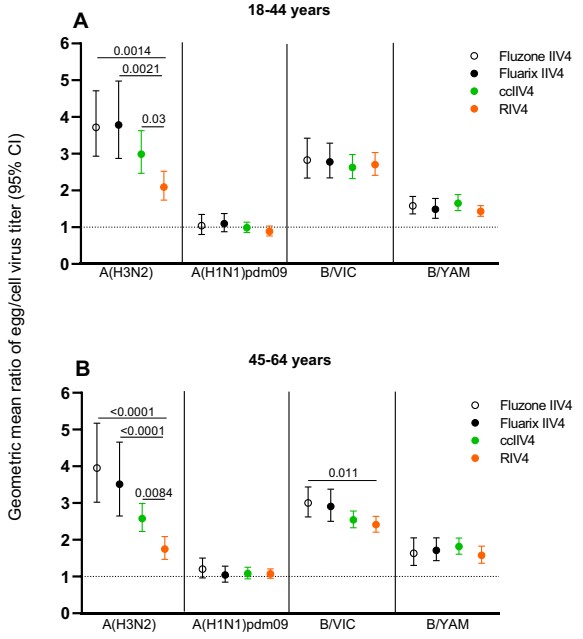

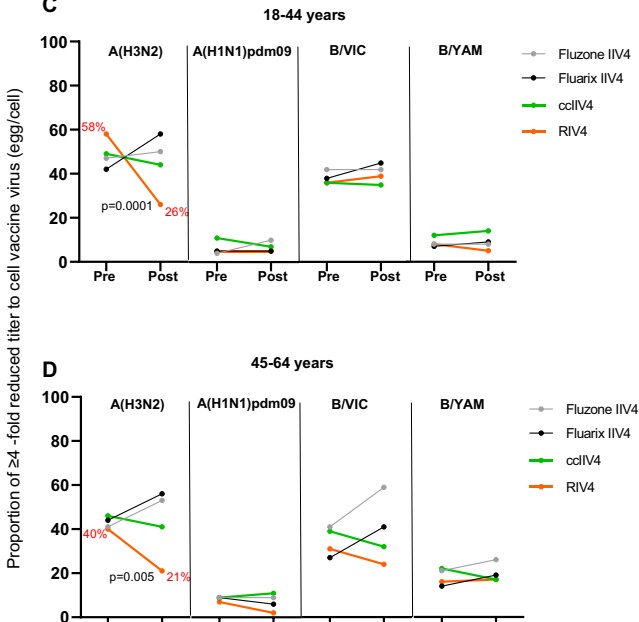

**Fig. 5 | Difference in MN or HI antibody responses to egg-adapted epitopes stratified by age groups and vaccine type in Year 1. A, B**: Geometric mean ratio of egg/cell virus titers at 1-month post-vaccination with 95% CI for each vaccine component among the 4 vaccine groups for 18-44 years (**A** Fluzone IIV4: $n = 52$, Fluarix IIV4: $n = 55$, ccIIV4: $n = 133$, RIV4: $n = 98$) and 45−64 years (**B** Fluzone IIV4: $n = 58$, Fluarix IIV4: $n = 63$, ccIIV4: $n = 147$, RIV4: $n = 98$) age groups. One-way ANOVA nonparametric Kruskal-Wallis test was used for comparison of geometric mean ratio of egg/cell virus titers among the 4 vaccine groups. **C, D**: Proportion of participants with GMT egg/cell ratio ≥4 fold pre- and 1-month post-vaccination in each vaccine group. Proportions of participants who had ≥4 -fold reduced titer to cell vaccine virus (egg/cell titer ratio) at pre- and 1-month post-vaccination were presented for each vaccine component among the 4 vaccine groups for 18−44 years (**C**) and 45−64 years (**D**). Fisher's exact test (two-tailed) was performed for comparing proportions of ≥4 -fold reduced titer to cell vaccine virus between pre- and 1-month post-vaccination in each vaccine group. MN titers were measured for A(H3N2) virus, while HI titers were measured for A(H1N1)pdm09, B/VIC, and B/YAM viruses. Statistically significant differences between groups are indicated by $p$ values on the horizontal bars (**A**, **B**). $P$ values on the trend lines of RIV4 group indicate significantly reduced proportions from pre- to 1-month post-vaccination (**C**, **D**).

A(H3N2) and B/VIC in the 18−44 years group, B/VIC virus and B/YAM virus in the 45−64 years group, suggesting repeated boost to antibodies targeting egg-adapted epitopes in these participants after receiving consecutive egg-based IIV4 vaccination. In contrast, among recipients of non-egg-based vaccines in just one study year (IIV4-ccIIV4 or IIV4-RIV4) or in both study years (ccIIV4-ccIIV4, ccIIV4-RIV4, RIV4-ccIIV4, RIV4-RIV4), the proportions of participants with egg/cell ratio ≥4 to generally decreased post-vaccination though did not reach statistical significance in some repeat vaccination groups (Fig. 7). Among those, the ccIIV4-RIV4 arm had significant ($p < 0.05$) reduction in the proportion of HI GMT egg/cell ratio ≥4 against A(H3N2) viruses comparing pre- vs post-vaccination in the 45−64 years group, and the RIV4-ccIIV4 arm had significant ($p < 0.05$) reduction in the proportion of HI GMT egg/cell ratio ≥4 against A(H1N1)pdm09 virus in the younger age group in year 2, suggesting multiple seasons of repeat vaccination with non-egg-based vaccines may be needed to overcome the dominant antibody responses to the egg-adapted epitopes and redirect the antibody responses away from the egg-adapted epitopes.

Total binding antibodies to HA head and HA stalk in year 2 were also analyzed (Fig. 8). At 1-month post-vaccination, the levels of total HA binding antibodies were similar in most repeat vaccination arms, except ccIIV4-RIV4 arm induced significantly higher ($p < 0.05$) HA head binding antibodies to A(H3N2), both ccIIV4-RIV4 and IIV4-RIV4 arms induced significantly higher ($p < 0.05$) HA head binding to B/Vic and B/Yam than the IIV4-IIV4 (Fluzone) arm (Fig. 8A). The total HA binding antibody GMT egg/cell ratio was around 1 against all 4 vaccine components with no significant difference between 7 repeat vaccination arms (Fig. 8B), indicating most of the total HA binding antibodies are not targeting the egg-adapted epitopes.

Comparing pre- and 1-month post-vaccination, A(H3N2) HA stalk binding antibodies were significantly boosted in ccIIV4-RIV4, RIV4-RIV4, IIV4-ccIIV4, and IIV4-RIV4 arms, while A(H1N1)pdm09 HA stalk binding antibodies were also significantly boosted in ccIIV4-ccIIV4 and RIV4-ccIIV4 arms (Fig. 8C). However, fold rise in both A(H1N1)pdm09 HA stalk and A(H3N2) HA stalk antibodies were low (GM fold rise <1.78) with no significant difference ($p > 0.05$) among all 7 repeat vaccination arms (Fig. 8D).

## Discussion

In this study, we compared the antibody responses to egg- versus cell-propagated vaccine viruses following vaccination with RIV4, ccIIV4 or IIV4 among HCP aged 18−44 and 45−64 years. RIV4 induced more robust HI and MN antibody responses than ccIIV4 and egg-based IIV4 against multiple vaccine strains including cell-propagated A(H3N2), consistent with previous reports[9,13,15,27−29]. Our analysis also indicates that repeat vaccination with non-egg-based vaccines could overcome pre-existing egg/cell titer differences in neutralizing antibody levels by re-directing vaccine-induced neutralizing antibody responses away from egg-adapted epitopes, resulting in higher antibody responses to circulating cell-grown viruses, even with frequent prior vaccination with egg-based influenza vaccines in HCP.

Reported VE against A(H3N2) virus was low during the 2016−17 to 2018−19 seasons despite the absence of apparent antigenic drift of the circulating viruses[1,4,31]. The egg-adapted changes on the HA head domain in the egg-grown A(H3N2) vaccine virus: T160K, L194P, and D225G resulted in an "antigenic mismatch" from circulating strains that likely contributed to the low observed VE[3,5,6]. Our study showed that even at pre-vaccination, a high proportion of HCP in this study had MN GMT egg/cell ratio ≥4 to A(H3N2) virus, suggesting the high prevalence

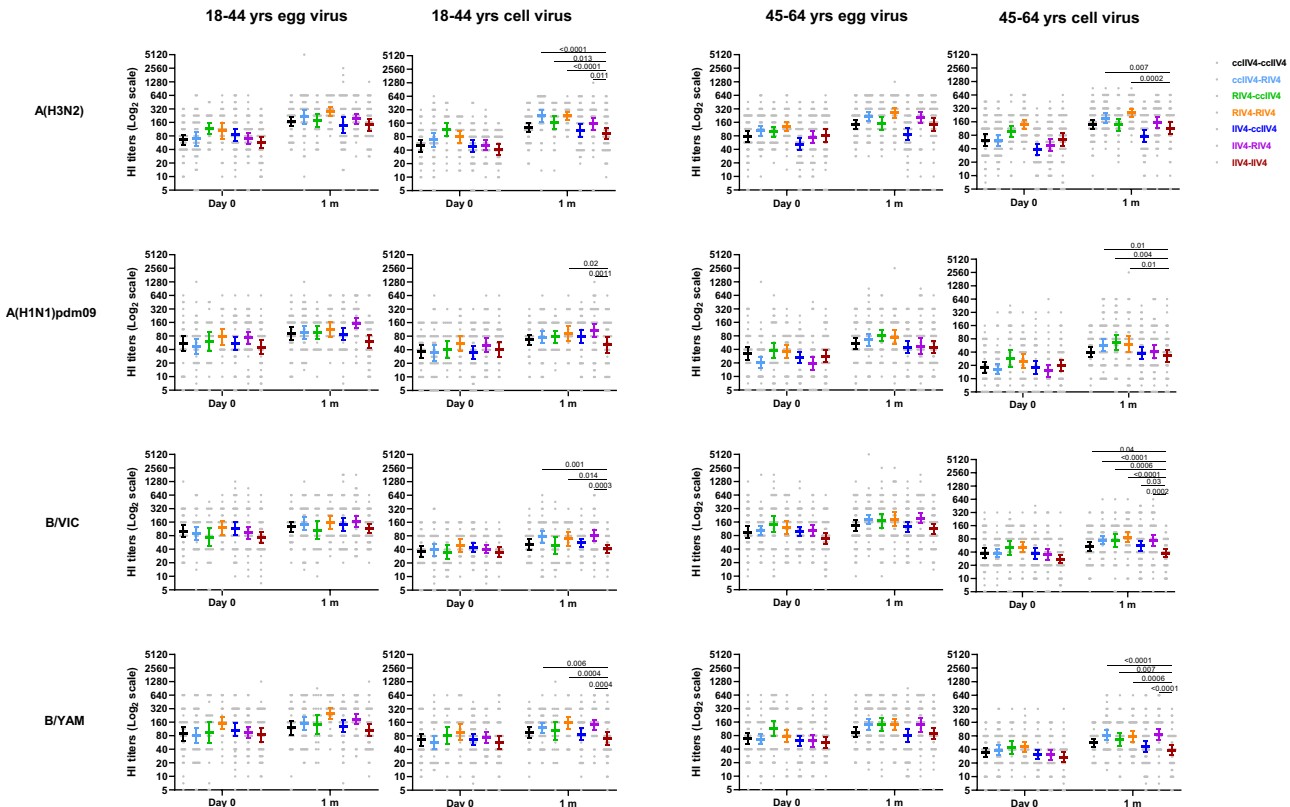

**Fig. 6 | HI antibody responses to egg- and cell-propagated vaccine viruses at pre- and 1-month post-vaccination in year 2 stratified by 7 repeat vaccination arms.** HI antibody titers from each of the 7 repeat vaccine arms were presented as geometric mean titers (GMTs) with 95% confidence interval (CI) respectively for 18–44 years and 45–64 years age groups. Gray dots represented individual titers. 18–44 years: ccIIV4-ccIIV4 (*n* = 42), ccIIV4-RIV4 (*n* = 45), RIV4-ccIIV4 (*n* = 28), RIV4-RIV4 (*n* = 30), IIV4-ccIIV4 (*n* = 49), IIV4-RIV4 (*n* = 48), IIV4-IIV4

(*n* = 47). 45–64 years: ccIIV4-ccIIV4 (*n* = 59), ccIIV4-RIV4 (*n* = 61), RIV4-ccIIV4 (*n* = 45), RIV4-RIV4 (*n* = 44), IIV4-ccIIV4 (*n* = 57), IIV4-RIV4 (*n* = 56), IIV4-IIV4 (*n* = 56). Unpaired *t* test (two-tailed) was used to compare the post-vaccination GMTs to cell vaccine viruses between IIV4-IIV4 (Fluzone) arm and each of the remaining 6 vaccine arms within the same age group. Statistically significant differences between groups are indicated by *p* values on the horizontal bars.

of pre-existing neutralizing antibodies targeting 158 (loss of glycosylation caused by T160K) 194P, and 225G epitopes, likely due to the prior repeat vaccination with egg-based vaccines. More than 68% of the HCP in this study received influenza egg-based vaccine in all five preceding influenza seasons (Table 2). The imprinting effect of egg-based vaccines could be lifelong resulting in a persistence of antibodies targeting egg-adapted sites that could be preferentially reinforced following repeat egg-based vaccination[3]. This may impede the quality of antibody response to the circulating influenza strains and negatively affect VE. Our study suggests that non-egg-based vaccines, especially RIV4, might circumvent the effect of repeat vaccination with egg-based influenza vaccines that contain egg-adapted changes.

Repeat vaccination can lead to reduced VE[23,32,33]. In year 1, the frequent prior repeat egg-based vaccinations may blunt the antibody responses to the subsequent vaccination as reflected by the limited fold rise in MN/HI antibody titers (Fig. 3). Khurana et al. also observed significantly lower fold rise to A(H3N2), A(H1N1) and influenza B viruses in those with prior vaccination irrespective of vaccine platforms received[20]. In a highly vaccinated population such as HCP, an effective vaccine should be able to overcome issues with both repeat vaccination and vaccine egg-adapted mutations. RIV4 vaccination reduced the proportions of participants with MN GMT egg/cell ratio ≥4 against A(H3N2) by two-fold in both age groups (Fig. 5C, D). In year 2, all 4 repeat vaccination arms with non-egg-based vaccines (ccIIV-ccIIV, ccIIV-RIV4, RIV4-ccIIV, RIV4-RIV4) further reduced the proportion of participants with HI GMT egg/cell ratio ≥4 against A(H3N2) to less than 10% (Fig. 7), of which 3 in 18–44 years group (ccIIV4-RIV4, RIV4-ccIIV4, RIV4-RIV4) and 2 in 45–64 years group (ccIIV4-RIV4, RIV4-RIV4)

induced significantly higher HI antibodies to A(H3N2) cell virus than IIV4-IIV4 arm (Fig. 6). Collectively, these findings suggest that multiple seasons of vaccination with non-egg-based vaccines may be needed to overcome the pre-existing immune memory to egg-adapted epitopes and re-direct the neutralizing antibody responses towards epitopes on cell-grown viruses that better represent circulating strains.

Furthermore, the difference in antigenicity and egg-adaption between vaccine viruses from consecutive influenza seasons could further confound the effect of repeat vaccination. The A(H3N2) vaccine virus in year 2 was updated to a 3 C.3a A/Kansas/14/2017-like virus, which is genetically and antigenically distant from the 3 C.2a vaccine virus A/Singapore/INFIMH-16-0019/2016 in year 1. Year 2 vaccine virus A/Kansas/14/2017 virus bears different egg-adapted changes at D190N and N246T but possess 194 L and 225D that are the same as cell-propagated A(H3N2) vaccine virus in Year 1 (Table 1). This could have also contributed to the improved antibody response to the cell-grown A(H3N2) vaccine virus after the repeat vaccination in year 2. Thus far, most reported VE from repeated vaccination studies were from egg-based vaccines, evaluation of whether repeat vaccination with non-egg-based vaccines will improve VE, especially for A(H3N2), is warranted.

To fully characterize the quantity as well as the quality of the antibody responses, we measured both functional, neutralizing antibody responses to egg- versus cell- propagated viruses (by MN or HI), and the total HA binding antibodies (by ELISA) that include both neutralizing and non-neutralizing antibody responses. For RIV4, both increased antigen dose and the absence of egg-adapted mutations may have led to the improved immunogenicity[13]. RIV4 contains 3 times the antigen dose at 45 μg HA/dose/strain compared to the standard dose

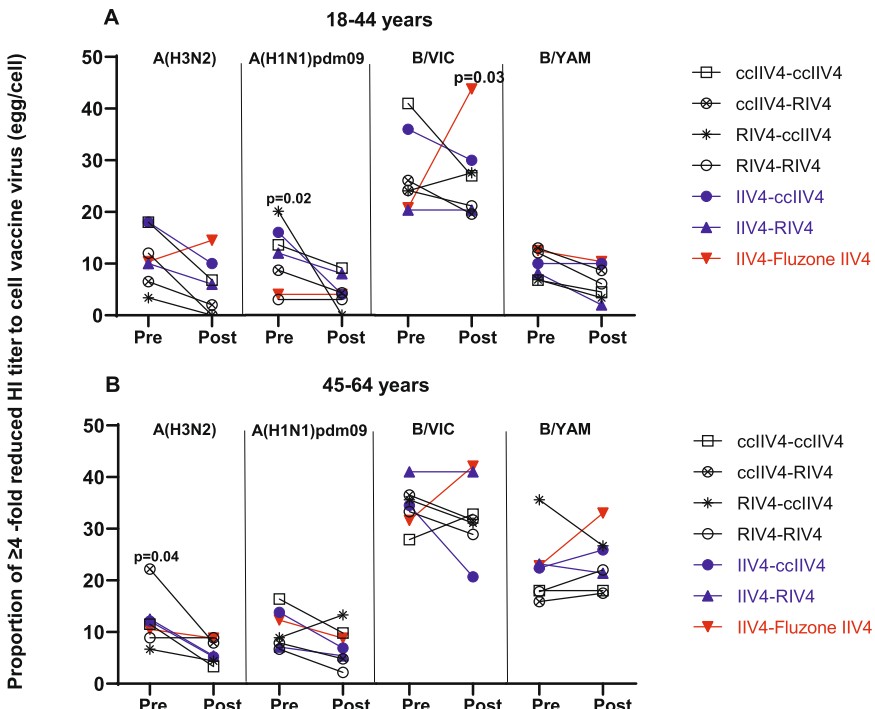

**Fig. 7 | Dynamics of proportion of participants with HI GMT egg/cell ratio ≥ 4 fold among seven repeat vaccination arms in Year 2.** Proportions of Year 2 participants who had ≥4-fold reduced titer to cell vaccine virus (egg/cell titer ratio) at pre- and 1-month post-vaccination were presented for each vaccine component among the seven Year 1-Year 2 vaccine arms. **A**: 18-44 years age group and **B**: 45−64 years age group. Fisher's exact test (two-tailed) was performed for comparing proportions of ≥4-fold reduced titer to cell vaccine virus between pre- and 1-month post-vaccination in each vaccine group. *P* values noted on the corresponding trend lines indicate statistically significant difference of the proportions between pre- and post-vaccination.

IIV4 and ccIIV4 (15 µg HA/dose/strain). For A(H3N2), RIV4 vaccination not only improved the quality of the functional neutralizing antibody responses (Fig. 2) by reducing the MN GMT egg/cell ratios in both years, it also significantly increased the quantity of the total A(H3N2) cell HA head binding antibodies (Fig. 4A, Fig. S1A). However, for A(H1N1)pdm09 and B viruses, the difference in HI antibody versus the total HA head binding antibody levels among 4 vaccine groups was less consistent, suggesting that the magnitude of improvement differ between subtypes and were likely determined by the relative immunodominance of the egg-adapted HA epitopes for each subtype. In addition, the egg/cell HA head binding ELISA titer ratios were around 1 for all vaccine antigen subtypes irrespective of vaccine platforms (Fig. 4B), suggesting the total HA binding antibody responses induced by vaccination cannot differentiate egg-adapted epitopes on HA even for A(H3N2). This is in stark contrast to the highly elevated egg/cell GMT ratio in neutralizing antibody responses to egg- vs cell-A(H3N2) viruses (Fig. 5A, B), highlighting the importance of inducing high quality, neutralizing antibody responses from vaccination.

Our study also demonstrated that the impact of vaccine egg-adaption may differ between influenza virus subtypes, with the biggest impact on neutralizing antibodies to A(H3N2) and moderate impact for influenza B vaccine viruses. HA changes due to egg-adaptation resulted in a loss of glycosylation at HA position 197 for both B/Victoria and B/Yamagata egg vaccine viruses (Table 1), which could alter the antigenicity of influenza B virus, particularly for B/Victoria lineage[34−36]. Disproportional impact of the egg-adapted mutation at HA position 197 on the two B lineages were also observed in this study. B/Victoria virus had a much higher GMT egg/cell titer ratio and proportion of participants with GMT egg/cell ≥4 than B/Yamagata virus in year 1 (Fig. 5). Similar to what was observed for A(H3N2), in Year 2, repeat vaccination with IIV4 boosted the antibody responses to egg-adapted epitopes at HA 197 in B/Victoria vaccine virus, resulting in substantially increased proportions of GMT egg/cell ratio ≥4 only in the IIV4-IIV4 arm (Fig. 7).

The impact of egg-adaptation was the least profound on the antibody response to A(H1N1)pdm09. Both Year 1 and Year 2 A(H1N1) pdm09 vaccine viruses bear egg-adapted change at HA Q223R. Studies have reported that Q223R egg-adapted change at the HA head domain of A(H1N1)pdm09 virus can promote virus replication in eggs, alter antigenicity and influence immune response[37−42]. In this study, 1-month post-vaccination GMTs to both egg- and cell-propagated A(H1N1) pdm09 vaccine viruses in year 1 were similar among the 4 vaccine groups, with GMT egg/cell titer ratio around 1.0, and the proportion of HCP with egg/cell titer ratio ≥4 was only 2−11% (Fig. 5), indicating a less immunodominant role of Q223R change. Furthermore, the two egg IIV4 vaccines induced similar antibody responses to A(H1N1)pdm09 viruses regardless of the variation at 223. These findings are consistent with our previous report in which we found that among over 300 adults who received egg-based A(H1N1)pdm09 vaccines with Q223R egg-adapted epitope during 5 influenza seasons, only 9% of participants mounted antibodies specific to 223R[38].

Multiple immune mechanisms can contribute to the protection from influenza infection. Broadly cross-reactive HA stalk antibodies have been considered as one of the targets for universal vaccine development. Here, we assessed both A(H1N1)pdm09 and A(H3N2) HA stalk antibody responses in all vaccine groups in both years. When comparing the post-vs pre-vaccination HA stalk antibody levels, vaccination induced HA stalk antibody rise in several vaccine groups and repeat vaccination arms. However, the levels of geometric mean titer fold rise were low (close to 1, Figs. 4D, 8D) with no significant difference in fold rise among different vaccine groups in both years, even in RIV4 group that received 3 times more antigen doses; the difference in HA stalk antibody titers post-vaccination mostly mirrored the difference in pre-existing HA stalk antibody levels. To this end, development of next generations vaccines that can induce multiple arms of immune responses could improve the effectiveness of the vaccines.

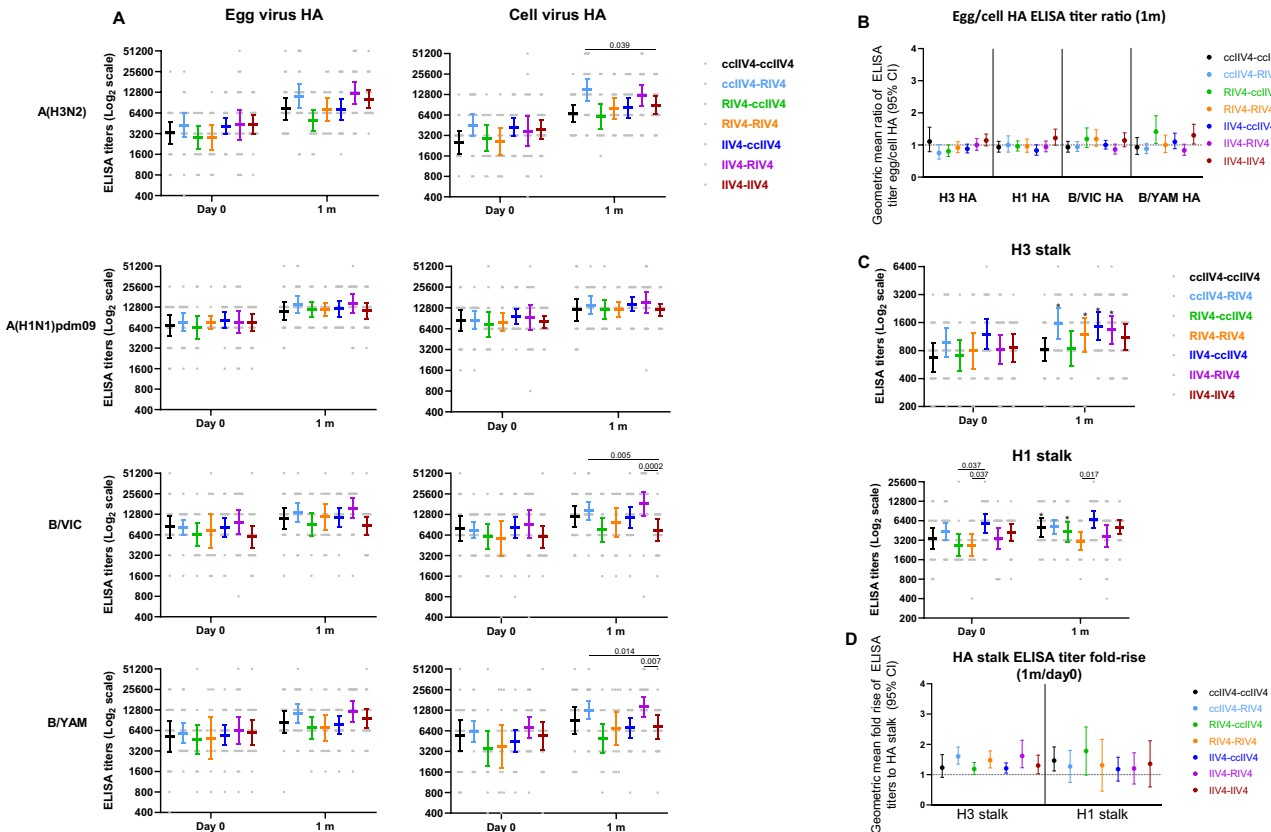

**Fig. 8 | HA head and stalk binding antibody responses measured by ELISA at pre- and 1-month post-vaccination in 7 repeat vaccination arms.** Antibody titers from each of the 7 vaccine arms were presented as geometric mean titers (GMTs) with 95% confidence interval (CI) for combined 18−44 years and 45−64 years age groups. Gray dots represented individual titers. ccIIV4-ccIIV4 ($n = 20$), ccIIV4-RIV4 ($n = 22$), RIV4-ccIIV4 ($n = 16$), RIV4-RIV4 ($n = 16$), IIV4-ccIIV4 ($n = 22$), IIV4-RIV4 ($n = 22$), IIV4-IIV4 ($n = 22$). **A:** HA head binding antibodies. **B:** Ratios of Egg/cell HA head binding antibody titers expressed as GM fold change with 95% CI. **C:** HA stalk binding antibodies for H3 and H1 stalk respectively. **D:** Fold-rise of HA stalk binding antibody titers from pre to 1 m post-vaccination expressed as GMFR with 95% CI. Unpaired *t* test (two-tailed) was used to compare the post-vaccination GMTs to cell virus HA between IIV4-IIV4 (Fluzone) arm and each of the remaining 6 vaccine arms

within the same age group (**A**). One-way ANOVA corrected for multiple comparisons (Tukey's test) was used to compare the GMTs of stalk binding antibody at each time point among the 4 vaccine groups (**C**). One-way ANOVA nonparametric Kruskal-Wallis test was used to compare the fold changes among the 4 vaccine groups (**B**, **D**). Statistically significant differences between groups are indicated by *p* values on the horizontal bars (**A**, **C**). *In (**C**), paired *t* test (two-tailed) was used for comparing the pre- and 1 m post-vaccination stalk antibody titers within the same vaccine group: significantly higher HA stalk antibody titers were detected in 1 m post-vaccination than pre-vaccination for H3 stalk antibody titers in ccIIV4-RIV4 ($p = 0.0001$), RIV4-RIV4 ($p = 0.004$), IIV4-ccIIV4 ($p = 0.03$), IIV4-RIV4 ($p = 0.03$); and for H1 stalk antibodies in ccIIV4-ccIIV4 ($p = 0.008$), RIV4-ccIIV4 ($p = 0.006$).

Human immunity to influenza is quite complex and present challenges to improve vaccine effectiveness. Immune priming can play dual roles and is largely directed by the immunodominance of imprinted HA epitopes upon first exposure to influenza antigen, as well as the repeated influenza exposures by vaccinations or infections later in lifetime[3,38]. Here, our study demonstrated how immunodominant egg-adapted epitopes on A(H3N2) HA can impact the immunogenicity from repeated vaccination. Collectively with previous reports[3], findings from this study could provide a few meaningful insights to improve vaccination strategy. First, for those who were repeat vaccinated with egg-based vaccines such as HCP and older adults, vaccination with non-egg-based vaccines could be especially beneficial; second, it would also be important not to prime naive children with egg-adapted epitopes[3], it might be necessary to immunize naive children with non-egg-based influenza vaccines as their first priming dose. In addition, improved vaccine formulations, such as higher vaccine doses and the use of adjuvants, could also improve vaccine immunogenicity.

Our study has several limitations. First, we only compared the immunogenicity of egg- vs non-egg-based vaccines in two influenza seasons. The effect of the vaccine egg-adapted changes on vaccine immunogenicity will vary by season depending on the antigenic properties of the vaccine viruses. A(H3N2) vaccine was updated to a

3 C.2a virus in 2021−2022 influenza season with the same T160K egg-adapted change in the egg-based vaccines and was updated again in 2022−23 and 2023−24 seasons with the egg-based vaccines carrying a different set of egg-adapted mutations[30], thus, continued effort is needed to assess vaccine immunogenicity and VE. Second, we only focused on the antibody responses to HA. RIV4 lacks the neuraminidase (NA) component that also could contribute to the protective efficacy of influenza vaccines. Thus, whether the improved antibody responses to HA alone in RIV4 will translate into improved vaccine efficacy and effectiveness still needs to be assessed. Other humoral immune responses, such as antibody responses to NA, mucosal and cell-mediated immunity, also contribute to protection. Lastly, no correction for multiple comparisons was done because this was an exploratory analysis.

Taken together, our study demonstrated that multiple seasons of vaccination with non-egg-based influenza vaccines can improve HA-mediated antibody immunogenicity to cell-propagated viruses representing circulating strains. Our findings support the use of antibody responses to cell-propagated viruses, rather than to egg-propagated vaccine viruses, as the primary endpoint in future immunogenicity studies for licensure or comparison of influenza vaccines. Influenza product and platform specific VE estimates could help confirm

**Table 2 | Participant characteristics in year 1**

| | Fluzone IIV4 | | Fluarix IIV4 | | ccIIV4 | | RIV4 | |
|---|---|---|---|---|---|---|---|---|
| | n = 122 | | n = 120 | | n = 283 | | n = 198 | |
| | n | % | n | % | n | % | n | % |
| Demographic characteristics | | | | | | | | |
| Mean age, years (SD) | 43 (11) | | 45 (11) | | 44 (11) | | 43 (12) | |
| Age Group | | | | | | | | |
| 18–44 years | 57 | 47 | 55 | 46 | 135 | 48 | 99 | 50 |
| 45–64 years | 65 | 53 | 65 | 54 | 148 | 52 | 99 | 50 |
| Female | 107 | 88 | 103 | 86 | 233 | 82 | 157 | 79 |
| White | 110 | 90 | 94 | 78 | 232 | 82 | 148 | 75 |
| Hispanic | 13 | 11 | 19 | 16 | 37 | 13 | 37 | 19 |
| Site | | | | | | | | |
| BSWH | 73 | 60 | 75 | 63 | 147 | 52 | 147 | 74 |
| KPNW | 49 | 40 | 45 | 37 | 136 | 48 | 51 | 26 |
| Prior influenza vaccination history | | | | | | | | |
| Receipt of influenza vacicnes in all the last 5 seasons | 83 | 68 | 93 | 78 | 209 | 74 | 135 | 68 |
| Prior season vaccination(2017–2018) | 108 | 89 | 104 | 87 | 245 | 87 | 176 | 89 |

findings from comparative immunogenicity studies and provide additional information to identify optimal vaccination strategies to prevent influenza.

## Materials and methods
### Study design
The original study was a randomized, open-label trial conducted in the United States during the Northern Hemisphere 2018–2019 (Year 1) and 2019–2020 (Year 2) influenza seasons. This study is an exploratory analysis. The detailed participant enrollment, randomization, and blinding were previously described[27–29]. Participant characterisitcs in year 1 are described in Table 2. During Year 1, HCP were stratified by two age groups (18–44 years and 45–64 years) and randomized at 4:4:2:2 ratio to receive one of four quadrivalent vaccines: ccIIV4 (Flucelvax), RIV4 (Flublok), Fluzone IIV4 or Fluarix IIV4 (Fig. 1). Serum samples were collected at baseline (day 0 pre-vaccination), 1-month, and 6-months post-vaccination. The Year 2 study was designed to evaluate the antibody responses following 7 different Year 1-Year 2 repeat vaccination regimens: IIV4-IIV4 (Fluzone), IIV4-ccIIV4, IIV4-RIV4, RIV4-ccIIV4, RIV4-RIV4, ccIIV4-ccIIV4 and ccIIV4-RIV4 (Fig. 1). Serum samples were collected at baseline (day 0) and 1-month post-vaccination.

Fluzone IIV4 and Fluarix IIV4, the two egg-based vaccines were analyzed separately for this exploratory analysis because the two vaccines have different egg-adapted mutation on the HA of the A(H1N1)pdm09 egg CVV (Table 1). Fluzone has Q223R substitution which was associated with focused antibody response to 223R in some vaccinees as we previously reported[38], while the A(H1N1)pdm09 egg CVV in Fluarix had 223Q. In addition, the current analysis is also age-stratified by 18–44 years and 45–64 years age groups as we previously observed that the impact of vaccine egg-adaptation on antibody response to A(H3N2) virus may be age-related[3].

### Hemagglutination inhibition (HI) and microneutralization (MN) assays
Antibody responses to A(H1N1)pdm09, B/Victoria and B/Yamagata lineage viruses were analyzed using HI assays using methods as previously described[43]. HIs for A(H1N1)pdm09 and influenza B viruses were performed using 0.5% turkey erythrocytes. Egg-grown viruses

were propagated in 10-day-old embryonated chicken eggs. Cell-grown A(H1N1)pdm09 and influenza B viruses were propagated in Madin-Darby canine kidney (MDCK) cells. Influenza B antigens were ether-treated prior to use in the HI assays. Egg- and cell (MDCK)-propagated A/Michigan/45/2015 (H1N1)pdm09, B/Colorado/06/2017 (B/Victoria), and B/Phuket/3073/2013 (B/Yamagata) viruses were tested in year 1, Egg-propagated A/Brisbane/02/2018 (H1N1)pdm09, cell-propagated A/Idaho/7/2018 (H1N1)pdm09, egg- and cell-propagated B/Colorado/06/2017 (B/Victoria) and B/Phuket/3073/2013 (B/Yamagata) viruses were tested in year 2 by HI using 0.5% turkey erythrocytes.

Antibody responses to A(H3N2) viruses were analyzed by MN assays in year 1 (the cell-grown vaccine antigen of the study year does not hemagglutinate in the HI assays), and HI assays in year 2. Egg-grown A(H3N2) viruses were propagated in 10-day-old embryonated chicken eggs, cell-grown A(H3N2) viruses were propagated in stably transfected MDCK cells with cDNA of human 2,6-sialyltransferase (MDCK-SIAT1 cells). Egg- and cell (MDCK-SIAT1)-grown A/Singapore/INFIMH-16-0019/2016 viruses were tested in MN assays in year 1 using MDCK-SIAT1 cells using methods as previously described[44]. In Year 2, egg- and cell-propagated A/Kansas/14/2017 viruses were tested by HI assays using 0.75% guinea pig erythrocytes in the presence of 20 nM oseltamivir.

### Enzyme-linked immunosorbent assay (ELISA)
Twenty percent of participants were selected by random sampling from each vaccine group and age strata (18–44 yrs and 45–64 yrs) in year 1 (n = 144) and year 2 (n = 140), the MN/HI antibody levels of the subset are representative of the whole study populations in year 1 (Fig. S3) and year 2 (Fig. S4). These serum samples were tested by ELISA to evaluate the total binding antibodies against 8 recombinant HA proteins (rHA from both egg-and cell-propagated viruses of each of the 4 vaccine antigens), and 2 rHA stalk proteins (A(H1N1)pdm09 HA stalk, and A(H3N2) HA stalk). rHA from egg- and cell A/Michigan/45/2015 (H1N1)pdm09, rHA head from egg- and cell- A/Singapore/INFIMH-16-0019/2016 (H3N2), B/Colorado/06/2017 (B/Victoria), and B/Phuket/3073/2013 (B/Yamagata) were tested with year 1 sera; rHA from egg- A/Brisbane/02/2018 (H1N1)pdm09 and cell-A/Idaho/7/2018 (H1N1)pdm09, rHA head from egg- and cell- A/Kansas/14/2017 (H3N2), egg- and cell- B/Colorado/06/2017 (B/Victoria), egg- and cell-B/Phuket/3073/2013 (B/Yamagata) were tested with year 2 sera. The purified and trimeric recombinant HA proteins were expressed from the baculovirus system using the established procedures[45]. For ELISA, rHA antigens were coated at 100 ng/well. Serum samples were tested at an initial dilution of 1:400 followed by 2-fold serial dilution. Antibodies were detected by horseradish peroxidase (HRP)-conjugated goat anti-human pan immunoglobulin conjugate to detect total binding antibodies. ELISA was performed based on previously described procedures[46,47]. The ELISA titers were determined as the reciprocal of the highest dilution of serum samples that achieved an optical density (OD) value of 0.2 or greater.

### Sequence analyses
All the testing viruses used in this study were sequenced. Sequences were analyzed against the sequences of candidate vaccine viruses (CVV) of the study vaccines as deposited in Global Initiative on Sharing Avian Influenza Data (GISAID) database by BioEdit version 7.0.9.0 (Table 1).

### Statistical analyses
Geometric mean titers (GMTs) at day 0, 1-month and 6-months, geometric mean fold rise (GMFR) of antibody titers at 1-month versus day 0, and geometric mean fold reduction of antibody titers at 6 months versus 1-month were calculated per vaccine group. "GMT egg/cell titer ratio" (ratio = GMT to egg-grown vaccine virus or rHA/GMT to cell-grown vaccine virus or rHA) was calculated to quantify the difference in antibody response to egg- versus corresponding cell-propagated vaccine virus for each vaccine group. The proportion of participants

with MN or HI GMT egg/cell titer ratio ≥ 4-fold was calculated to quantify the percent of participants with antibodies predominantly targeting egg-adapted epitopes per vaccine group. One-way ANOVA corrected for multiple comparisons (Tukey's test) or nonparametric Kruskal-Wallis test was used for multiple group comparison, two-tailed paired *t* test was used to compare pre- and post-vaccination; two-tailed unpaired *t* test was used to compare the post-vaccination titers between IIV4-IIV4 (Fluzone) arm and each of the remaining 6 arms in Year 2; Two-tailed Fisher's exact test was used for comparing proportions. $P < 0.05$ is considered statistically significant. SAS 9.4 (SAS Institute) and GraphPad Prism 8 (GraphPad Software, Inc.) were used for statistical analyses.

### Ethic review
The study protocol was reviewed and approved by the institutional review boards (IRBs) of the study sites and the Centers for Disease Control and Prevention. Informed consent was obtained from the participants. This study is registered on ClinicalTrials.gov, NCT03722589.

### Reporting summary
Further information on research design is available in the Nature Portfolio Reporting Summary linked to this article.

## Data availability
The data that support the findings of this study are included in the manuscript, supplementary information, and the figshare repository https://doi.org/10.6084/m9.figshare.22666231. Additional raw data from the study are available from the corresponding author (mlevine@cdc.gov) upon request.

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

## Acknowledgements

We thank the staff from Baylor Scott & White Health and Kaiser Permanente Northwest for their participation and support of the study. We thank Dr James Stevens and Paul Carney from the influenza division, CDC for providing the recombinant HA proteins. We also thank Makeda Kay and Stacie Jefferson from influenza division of CDC for their assistance with the study. Funding: This study was funded by the US Centers for Disease Control and Prevention (contract 75D30118F02850). Disclaimer: The findings and conclusions in this report are those of the authors and do not necessarily represent the views of the US Centers for Disease Control and Prevention.

## Author contributions

M.Z.L., F.S.D., B.F., A.M.F., M.G.T., T.T., M.G. and A.L.N. were involved in the conception and the design of the studies. F.L., F.L.G., S.J., K.M., H.G., M.G.W., L.J.E., L.G., S.S.K., S.S., S.G., performed the studies and acquired the data. F.L. and S.J. analyzed the data. F.L. and M.Z.L. wrote the manuscript. M.Z.L. supervised the study. All co-authors were involved in manuscript preparation process for important intellectual content.

## Competing interests

M.G. received funding from Pfizer for an unrelated educational grant and Janssen for an unrelated study. A.L.N. received research funding from Pfizer and Vir Biotechnology for unrelated studies. All other authors declared no conflict of interest.

## Additional information

[1]Influenza Division, Centers for Disease Control and Prevention, Atlanta, GA, USA. [2]Baylor Scott & White Health, Temple, TX, USA. [3]Baylor College of Medicine, Temple, TX, USA. [4]Texas A & M University, College of Medicine, Temple, TX, USA. [5]Kaiser Permanente Northwest Center for Health Research, Portland, OR, USA. [6]Abt Associates, Atlanta, GA, USA. ✉e-mail: mlevine@cdc.gov

