## [Peer Review File · Nature Communications]

Redirecting antibody responses from egg-adapted epitopes following repeat vaccination with recombinant or cell culture-based versus egg-based influenza vaccinesReviewers' Comments:

Reviewer #1:

Remarks to the Author:

Liu et al performed an interesting study to understand the impact of repeat vaccination with different vaccine platforms on responses to egg and cell-derived influenza viruses.

While the topic is timely and requires better understanding, the authors are missing some important information about impact of repeat vaccinations on antibody response quality and its relationship with vaccine effectiveness in this study. Even though numbers in each arm are small, can authors add the information on Flu infection or disease in this cohort that was followed for upto 6 months in each of the 2 years. This will help the field to makes sense of clinical relevance of the findings in the study and the next steps for more effective influenza vaccine approach.

Major Comments:

1. While the topic is timely and requires better understanding, the authors are missing some important information about impact of repeat vaccinations on antibody response and relevance of them to the Vaccine efficacy findings.
2. Since HAI titer of 40 or more is considered at least one of the correlates of protection, do different vaccine platforms impact the seropositivity rates (HAI >40) in the 2 years in these adults? In the group comparisons, add seropositivity rates (HAI>40) for each group in a table or figure and discuss the findings.
3. The better HAI response induced by RIV is not surprising as it generated higher titers (possibly due to higher HA antigen contained in the RIV compared with CCIV and IIV) as observed by previous studies. But the difference in egg vs cell-based viruses require further evaluation of HA-binding antibodies (by ELISA etc). Does the better factorial immune response to cell vs egg for RIV is just due to higher HA binding antibodies induced by RIV overall or is their qualitative difference in induced antibodies against cell vs egg epitopes should be resolved further in this manuscript.
4. To follow up on comments above, it will be important to evaluate and compare the HA-binding antibodies by different vaccine platforms in both years against cell vs egg derived HA. This is critical because several studies have suggested a role for non-neutralizing HA binding antibodies in clinical outcome.
5. In Figure 2, why RIV induced antibody against H1N1 that are highest at 1-month among different vaccine groups, decline at faster rate by 6-month compared with other IIV or ccIIV against egg or cell viruses. Please discuss. This is puzzling. It will be important to confirm these findings in a binding assay (ELISA etc) to demonstrate that indeed the anti-HA antibody titers are declining at faster rate.
6. Moreover, why no increased immune response against B strains or cell following vaccination in any group, especially B Yam in older age group.
7. From figure 3, it seems that most vaccines generated only limited increase in HAI titers at 1-month post-vaccination over baseline. One of the criteria for evaluation of Flu vaccine immunogenicity is induction of 4-fold increase in HAI titers following vaccination. Authors should add seroconversion rates (table or figure) and discuss the findings in context of previous studies.
8. The role of egg vs cell-based vaccine in 2 years is interesting. What is the role of antigenic sin and how does it impact the immune response. Please discuss further with current literature and how to overcome with better vaccine approaches going forward?
9. Earlier studies and meta analysis have postulated that repeat vaccination given every year led to reduced antibody response, quality and vaccine efficacy. Can authors discuss if repeat vaccination (egg or cell) leading to reduced VE is possibly due to enhanced disease. Like what was speculated during the 2009 H1N1 pdm09 in Canada (Skowronski et al) or enhance influenza disease following mismatched H1N1pdm09 virus in animal models (PMID: 23986398).
10. Authors should also provide perspective on limitations and gaps that can help the move the field forward and clinical relevance of these findings for designing better vaccine approaches to address the key issue of improving vaccine efficacy against Influenza. Do these findings have any role in

development of next generation flu vaccines?

Minor Comment:

1. In results section, authors should add GMT titers for each group when describing the results for easy reading and comparisons.

Reviewer #2:

Remarks to the Author:

This is a terrific study from Liu and CDC colleagues characterizing human antibody responses following vaccination with egg-based, cell-based, and recombinant protein-based influenza vaccines. Consistent with previous studies, the authors show that the recombinant protein-based vaccines (which possess 'wild type' immunogens) elicit higher neutralizing antibody titers against 'wild type' viruses, especially H3N2. Importantly, the authors measure antibody responses over 2 consecutive seasons with participants receiving different cell/egg/recombinant protein vaccines. They find that vaccination with cell and recombinant protein vaccines can elicit high levels of antibodies against wild type viruses in individuals who previously received egg-based vaccines, and there seems to be an additive effect when cell and recombinant protein vaccines were delivered in consecutive years (in a population previously vaccinated many times with egg-based vaccines). The study is important—the authors nicely describe caveats of the experiments and how it is unknown how the serological data will relate to VE—however even with these caveats, it seems pretty clear to me that we should abandon egg-based influenza vaccines and increase our efforts to produce large amounts of recombinant protein based vaccines. This is a very important study that should shape public policy.

1. The different amount of antigen in each vaccine is discussed at the end, but this information should also be provided in the introduction for readers that might not know this.
2. It seems odd to present the egg/cell ratio data of Year 2 first before presenting Year 2 antibody data. Please consider switching the order of Figure 6 and 7.
3. Were experiments completed to measure antibody titers to Year 1 vaccine strains with Year 2 serum samples? It might be interesting to see if the antigenically distinct Year 2 strains boosted antibodies to the Year 1 viruses. Consider adding these data if these experiments have already been completed.
4. Line 177 on pg 8 should be revised because this doesn't appear to be the case with H3 responses in 45-64 year olds.

REVIEWER

COMMENTS

Response to Reviewer 1:

Reviewer #1 (Remarks to the Author):

Liu et al performed an interesting study to understand the impact of repeat vaccination with different vaccine platforms on responses to egg and cell-derived influenza viruses.

While the topic is timely and requires better understanding, the authors are missing some important information about impact of repeat vaccinations on antibody response quality and its relationship with vaccine effectiveness in this study. Even though numbers in each arm are small, can authors add the information on Flu infection or disease in this cohort that was followed for up to 6 months in each of the 2 years. This will help the field to makes sense of clinical relevance of the findings in the study and the next steps for more effective influenza vaccine approach.

Response:

Thanks for the comment. In this trial, a total of 723 participants were enrolled in year 1 and 682 participants were enrolled in year 2 which are substantial sample sizes for an immunogenicity study, but unfortunately not statistically powered to assess efficacy due to the low attack rate of influenza positive cases. Active surveillance was conducted in year 1 (not year 2) to identify influenza cases by PCR. There was a total 26 PCR confirmed influenza infection (vaccine breakthrough cases) in year 1 out of a total of 723 participants, resulting in an attack rate of only 3.6%. the vaccine breakthrough cases were identified from all vaccine groups. We added a summary of the break-through influenza infection cases from the Year 1 participants by vaccine types (Supplementary materials Table S3) that will provide additional information on the clinical relevance of the study finding in this manuscript. As we emphasized in the manuscript, continued efforts are needed to assess platform specific vaccine effectiveness (VE), including the VE for RIV4 from multiple influenza seasons. Furthermore, currently most VE reports from repeated vaccination studies are based on egg-based vaccines, more studies are needed to monitor VE from repeated vaccination with non-egg-based.

To address the reviewer's comments, we added the summary of vaccine breakthrough cases in the results, it reads:

“Active surveillance was conducted during the influenza season, a total of 26 breakthrough infection cases were identified in year 1 with an attack rate of 3.6% (26/723) (Supplementary Table S3). “

And we discussed the link of our finding with clinical relevance, and next steps in several places in the discussion, such as below:

“Collectively, these findings suggest that multiple seasons of vaccination with non-egg-based vaccines may be needed to overcome the pre-existing immune memory to egg-adapted epitopes and re-direct the neutralizing antibody responses towards epitopes on cell-grown viruses that better represent circulating strains.

“Thus far, most reported VE from repeated vaccination studies were from egg-based vaccines, evaluation of whether repeat vaccination with non-egg-based vaccines will improve VE, especially for A(H3N2), is warranted.”

Major Comments:

1. While the topic is timely and requires better understanding, the authors are missing some important information about impact of repeat vaccinations on antibody response and relevance of them to the Vaccine efficacy findings.

Response:

Thanks for the comments. We have included additional discussion on this point, three additional publications were also cited: Ohmit et al. 2014; Jones-Gray et al. 2023; Ramsay et al 2019.

We have updated the discussion in the main text as below, and believe we have addressed this comment adequately.

“Repeat vaccination can lead to reduced VE (Ohmit et al. 2014; Jones-Gray et al. 2023; Ramsay et al 2019). In Year 1, the frequent prior repeat egg-based vaccinations may blunt the antibody responses to the subsequent vaccination as reflected by the limited fold-rise in MN/HI antibody titers (Figure 3). Khurana et al. also observed significantly lower fold-rise to A(H3N2), A(H1N1) and influenza B viruses in those with prior vaccination irrespective of vaccine platforms received. 4 In a highly vaccinated population such as HCPs, an effective vaccine should be able to overcome issues with both repeated vaccination and vaccine egg-adapted mutations. RIV4 vaccination reduced the proportions of participants with MN GMT egg/cell ratio ≥ 4 against A(H3N2) by two-fold in both age groups (Figure 5C and 5D). In Year 2, all four repeat vaccination arms with non-egg-based vaccines (cclIV-cclIV4, cclIV4-RIV4, RIV4-cclIV4, RIV4-RIV4), further reduced the proportion of participants with HI GMT egg/cell ratio ≥ 4 against A(H3N2) to less than 10% (Figure 7), of which three in 18-44 yrs group (cclIV4-RIV4, RIV4-cclIV4, RIV4-RIV4) and two in 45-64 years group (cclIV4-RIV4, RIV4-RIV4) induced significantly higher HI antibodies to A(H3N2) cell virus than IIV4-IIV4 arm (Figure 6). Collectively, these findings suggest that multiple seasons of vaccination with non-egg-based vaccines may be needed to overcome the pre-existing immune memory to egg-adapted epitopes and re-direct the neutralizing antibody responses towards epitopes on cell-grown viruses that better represent circulating strains.

Furthermore, the difference in antigenicity and egg-adaption between vaccine viruses from consecutive influenza seasons could further confound the effect of repeat vaccination. The A(H3N2) vaccine virus in Year 2 was updated to a 3C.3a A/Kansas/14/2017-like virus, which is genetically and antigenically distant from the 3C.2a vaccine virus A/Singapore/INFIMH-16-0019/2016 in Year 1. Year 2 vaccine virus A/Kansas/14/2017 virus bears different egg-adapted changes at D190N and N246T but possess 194L and 225D that are the same as cell-propagated A(H3N2) vaccine virus in Year 1 (Table 1). This could have also contributed to the improved antibody response to the cell-grown A(H3N2) vaccine virus after the repeat vaccination in year 2. Thus far, most reported VE from repeated vaccination studies

were from egg-based vaccines, evaluation of whether repeat vaccination with non-egg-based vaccines will improve VE, especially for A(H3N2), is warranted. “

2. Since HAI titer of 40 or more is considered at least one of the correlates of protection, do different vaccine platforms impact the seropositivity rates (HAI >40) in the 2 years in these adults? In the group comparisons, add seropositivity rates (HAI>40) for each group in a table or figure and discuss the findings.

Response:

Thanks for the comments. To address the reviewer’s comments, we have now added 4 supplementary tables to summarize the seropositivity rates (HAI \geq 40) to both egg- and cell- grown viruses stratified by age groups for both Year 1 (Table S1 and S2) and Year 2 participants (Table S4 and S5). The seropositivity rates to cell-grown vaccine viruses (not to the egg-grown vaccine antigens and not stratified by age groups) have been published previously and referenced in this manuscript (Dawood et al. 2021; Gaglani et al. 2022; Naleway et al. 2023).

Overall, RIV4 demonstrated significantly higher post-vaccination antibody responses by seropositivity rate to cell-grown vaccine viruses compared to egg-based IIV4 vaccines. Repeat vaccination with non-egg-based vaccine cIIV4 or RIV4 in 2 seasons, or RIV4 in the second season also resulted in significantly higher post-vaccination seropositivity rate to cell-grown vaccine viruses than repeat vaccination with egg-based IIV4 in 2 consecutive seasons (IIV4-IIV4).

3. The better HAI response induced by RIV is not surprising as it generated higher titers (possibly due to higher HA antigen contained in the RIV compared with CCIV and IIV) as observed by previous studies. But the difference in egg vs cell-based viruses require further evaluation of HA-binding antibodies (by ELISA etc). Does the better factorial immune response to cell vs egg for RIV is just due to higher HA binding antibodies induced by RIV overall or is their qualitative difference in induced antibodies against cell vs egg epitopes should be resolved further in this manuscript.

Response:

The reviewer raised a very good point. We have invested significant efforts in the revision to thoroughly address this comment from the reviewer. We have performed additional experiments, run ELISA to both HA head and HA stalk from each subtype, tested against sera from all time points of each vaccine arms in both Year 1 and Year 2 of the study to evaluate the total HA binding antibodies. We have included new data from total HA head binding antibodies as well as H1 stalk and H3 stalk antibodies in this revision. Pre- and post-vaccination serum samples from pre, 1m, and 6m post-vaccination in year 1, sera from pre and 1m post-vaccination in year 2 were tested in ELISA to detect the pan Ig antibodies against 8 recombinant HA head and 2 HA stalk antigens from all 4 subtypes. These new data were analyzed and summarized in 4 new figures of the revised manuscript: Figure 4 (year 1), Figure S1&S2 (year 1) and Figure 8 (year 2).

For A(H3N2), RIV4 vaccination not only improved the quality of the functional neutralizing antibody responses (Figure 2) by reducing the MN GMT egg/cell ratios in both years, it also significantly increased the quantity of the total H3 cell HA head binding antibodies (Figure 4A, Figure S1A). However, for A(H1N1)pdm09 and B viruses, the difference in HI antibody versus the total HA head binding antibody levels among 4 vaccine groups was less consistent, suggesting that the magnitude of improvement differ between subtypes and were likely determined by the relative immunodominance of the egg-adapted HA epitopes for each subtype.

Furthermore, we have now included detailed results and discussion in the manuscript to describe the data from the binding antibodies from both years, and added additional points in the discussion.

Total binding antibody results from year 1 are described as below:

“To further elucidate the quality of the antibody responses, we then analyzed the total binding antibodies by the enzyme linked immunosorbent assay (ELISA) to both HA head and HA stalk of the 4 antigen subtypes in the quadrivalent vaccines. Total binding antibodies include both neutralizing and non-neutralizing antibody responses. For A(H3N2), consistent with the MN responses, RIV4 induced significantly higher ($p<0.01$) total binding antibodies (Figure 4A) and significantly higher ($p<0.001$) fold rise to the cell-A(H3N2) HA head (Figure S1A) at 1 month post-vaccination compared to IIV4 and cIIIV4. For A(H1N1)pdm09, all 4 vaccine induced similar fold rise (1 month/day 0) in the total binding antibodies to egg and cell- A(H1N1)pdm09 HA (Figure S1A), although RIV4 group had the highest total binding antibodies to the cell-A(H1N1)pdm09 HA1 at 1 month post-vaccination, it is likely due to the higher pre-vaccination total binding antibody levels (Figure 4A). For influenza Bs, RIV4 induced significantly higher ($p<0.01$) fold-rise against B/VIC and B/YAM cell virus HA than Fluarix-IIV4 but not Fluzone-IIV4 (Figure S1A). At 6 months post vaccination, total binding antibodies to HA head waned but the levels of total binding antibodies to the cell-grown virus HA head of all 4 subtypes in the RIV4 group remained the highest among the 4 vaccine groups (Figure 4A).”

“We then also analyzed the HA binding antibody “GMT egg/cell titer ratio” measured by ELISA. Similar trend was observed as with MN/HI antibodies that GMT egg/cell ratios were higher ($p<0.05$) in the egg-based IIV4 groups than cIIIV4 and RIV4 groups for A(H3N2) HA head and B/VIC HA head binding antibodies (Figure 4B). However, the GMT egg/cell binding antibody ratios to HA head (Figure 4B) were around 1 for all vaccine groups and subtypes, even for A(H3N2), suggesting that unlike the neutralizing antibody responses, the majority of total binding antibody responses do not target the egg-adapted epitopes.”

Total binding antibody responses from year 2 are described as below:

“Total binding antibodies to HA head and stalk in year 2 were also analyzed (Figure 8). At 1-month post-vaccination, the levels of total binding antibodies to all 4 subtypes HA head were similar in most repeat vaccination arms, except cIIIV4-RIV4 arm induced significantly higher ($p<0.05$) head binding antibodies to H3, B/Vic and B/yam than the IIV4-IIV4 (Fluzone) arm. The GMT egg/cell total HA binding antibody titer ratio was around 1 against all 4 vaccine components with no significant difference between 7 repeat vaccination arms (Figure 8B), indicating most of the total binding antibodies are not targeting the egg-adapted epitopes.”

And we included additional discussion on the role of binding antibodies:

“To fully characterize the quantity as well as the quality of the antibody responses, we measured both functional, neutralizing antibody responses to egg- versus cell- propagated viruses (by MN or HI), and the total HA binding antibodies (by ELISA) that includes both neutralizing and non-neutralizing antibody responses. For RIV4, both increased antigen dose and the absence of egg-adapted mutations may have led to the improved immunogenicity. RIV4 contains 3 times the antigen dose at 45ug HA/dose/strain compared to the standard dose IIV4 and cIIIV4 (15ug HA/dose/strain). For A(H3N2), RIV4 vaccination not only improved the quality of the functional neutralizing antibody responses (Figure 2) by reducing the MN GMT egg/cell ratios in both years, it also significantly increased the quantity of the total H3 cell HA head binding antibodies (Figure 4A, Figure S1A). However, for A(H1N1)pdm09 and B viruses, the difference in HI antibody versus the total HA head binding antibody levels among 4 vaccine groups was less consistent, suggesting that the magnitude of improvement differ between subtypes and were likely determined by the relative immunodominance of the egg-adapted HA epitopes for each subtype. In addition, the egg/cell HA head binding ELISA titer ratios were around 1 for all vaccine antigen subtypes irrespective of vaccine platforms (Figure 4B), suggesting the total binding antibody responses induced by vaccination cannot differentiate egg-adapted epitopes even for A(H3N2). This is in stark contrast to the highly elevated egg/cell GMT ratio in neutralizing antibody responses to egg- vs cell-A(H3N2) viruses (Figure 5A&B), highlighting the importance of inducing high quality, neutralizing antibody responses from vaccination. ”

4. To follow up on comments above, it will be important to evaluate and compare the HA-binding antibodies by different vaccine platforms in both years against cell vs egg derived HA. This is critical because several studies have suggested a role for non-neutralizing HA binding antibodies in clinical outcome.

Response:

Please see detailed response to above comment 3 on the HA total binding antibody responses, including new results and discussion on non-neutralizing antibodies included in this revision.

Furthermore, to fully elucidate the quality of the antibody responses, we also analyzed HA stalk antibody levels in all vaccine groups from both year 1 and year 2. The results are summarized in Figure 4C and 4D, and Figure 8C and 8D, and Figure S2.

We added HA stalk antibody responses results as below:

HA stalk antibody responses in year 1:

“Pre-existing H3 and H1 HA stalk antibodies were detected in all participants at varies levels. Comparing stalk antibody titers at 1-month versus day 0, vaccination significantly boosted H1 stalk antibodies in all 4 vaccine groups ($p < 0.05$), and significantly boosted H3 stalk binding antibody responses in 3 of the 4 vaccine groups ($p < 0.05$, Fluzone-IIV4, cIIIV4, and RIV4) (Figure 4C). The RIV4 group had the highest H1 stalk and H3 stalk antibodies at 1-month post-vaccination, however the fold rise of stalk antibodies was low (< 1.8) and similar among all 4 vaccine groups (Figure 4D). At 6-months post-vaccination, waning of stalk antibodies were notable in cIIIV4 and RIV4 groups compared to IIV4 (Figure S2).”

HA stalk antibody responses in year 2:

“H3 stalk antibodies were significantly boosted in cclIV4-RIV4, RIV4-RIV4, IIV4-cclIV4, and IIV4-RIV4 arms, while H1 stalk binding antibodies were also significantly boosted in cclIV4-cclIV4 and RIV4-cclIV4 arms comparing pre- and 1-month post-vaccination (Figure 8C). However, fold rises in both H1 stalk and H3 stalk antibodies were low (mean fold rise <1.78) with no significant difference ($p < 0.05$) among all 7 repeat vaccination arms (Figure 8D).”

We also added additional discussion:

“Multiple immune mechanisms can contribute to the protection from influenza infection. Broadly cross-reactive HA stalk antibodies have been considered as one of the targets for universal vaccine development. Here, we assessed both H1 and H3 HA stalk antibody responses in all vaccine groups in both years. When comparing the post-vs pre-vaccination HA stalk antibody levels, vaccination induced HA stalk antibody rise in several vaccine groups and repeated vaccination arms, however the levels of GM fold rise were low (close to 1, Figure 4 and Figure 8) with no significant difference in fold rise among different vaccine group in both years, even in RIV4 group that received 3 times more antigen doses. The difference in HA stalk antibody titers post-vaccination mostly reflect the difference in pre-existing stalk antibody levels. To this end, development of next generations vaccines that can induce multiple arms of immune responses could improve the effectiveness of the vaccines.”

5. In Figure 2, why RIV induced antibody against H1N1 that are highest at 1-month among different vaccine groups, decline at faster rate by 6-month compared with other IIV or cclIV against egg or cell viruses. Please discuss. This is puzzling. It will be important to confirm these findings in a binding assay (ELISA etc) to demonstrate that indeed the anti-HA antibody titers are declining at faster rate.

Response:

For antibody responses to A(H1N1)pdm09 at 1-month post-vaccination, in Figure 2, HI antibody titers in the RIV4 groups to cell-A(H1N1)pdm09 was only statistically significantly ($p < 0.05$) higher than the cclIV4 group in the 18-44 yrs group, and higher than the Fluzone-IIV4 group in the 45-65 yrs group, they were not statistically significantly higher ($p > 0.05$) than other vaccine groups (Figure 2). The high antigen doses in RIV4, at 45ug/antigen, 3 x more than the standard doses in IIV4 and cclIV4, could have contributed to the improved antibody responses. However, as we noted in the manuscript, the improvement in antibody responses is subtype dependent as illustrated in Figure 2, the highest increase of post-vaccination titers was for A(H3N2), and less so for A(H1N1)pdm09 and influenza Bs.

In terms of rate of antibody waning for A(H1N1)pdm09, 6M/1M fold reduction data can be found in Figure 3B. For A(H1N1)pdm09, RIV4 6 months antibody waning rate was similar ($p < 0.05$) to most vaccine groups in both age ranges, except in the 45-64 age groups, 6M/1M fold reduction is higher than the cclIV4 group only. Important to note is that at 6 months post-vaccination for A(H1N1)pdm09, the absolute antibody titers in the RIV4 group were still higher than or similar to other vaccine groups (Figure 2).

We also further investigated this point using the binding assay with new data included in Figure 4A and Supplementary figure S1B. ELISA titers overall had a similar trend as HI titers.

6. Moreover, why no increased immune response against B strains or cell following vaccination in any group, especially B Yam in older age group.

Response:

As shown in Figure 3A, there is minimal GM fold-rise from pre- to post-vaccination against B viruses. The GM fold-rise against both B/VIC and B/YAM viruses ranged from 1.2 to 1.9 with RIV4 mostly inducing the highest fold-rise.

There are two possible explanations:

1. These are HCPs with repeat vaccination from the prior 5 influenza seasons, prior vaccination can blunt the antibody responses. Many had high pre-vaccination antibody titers, and studies have shown that high pre-vaccination antibody titers can be inversely correlated with post-vaccination antibody fold rise.
2. Another plausible explanation is that there may be difference in immunogenicity of each of the 4 vaccine components and there are some antigen competitions in their ability to induce antibody responses. Our studies have shown that antibody responses to each of the 4 vaccine components in the quadrivalent vaccines are often at different levels. Here, competition between the HA antigens formulated into the quadrivalent influenza vaccines with H3 and H1 HA antigens may be more immunogenic than B HA antigens. A study led by Khurana et al. observed the similar trend: antibody response to B viruses was much lower compared to A(H3N2) and A(H1N1) viruses in both GMTs and fold-rise at 28 days post-vaccination, irrespective of vaccine types received.

This comment was also addressed collectively with our responses to comment 7 below.

7. From figure 3, it seems that most vaccines generated only limited increase in HAI titers at 1-month post-vaccination over baseline. One of the criteria for evaluation of Flu vaccine immunogenicity is induction of 4-fold increase in HAI titers following vaccination. Authors should add seroconversion rates (table or figure) and discuss the findings in context of previous studies.

Response:

Thanks for the comments. As stated in our responses to comment 2, the seroconversion rate (% of 4-fold rise) of both study years to both egg- and cell- vaccine viruses stratified by age groups are now included in additional supplementary tables for both year 1 (Supplementary Table S2) and year 2 (Supplementary Table S5). We also referenced previous publications on seropositivity rates to the cell-grown viruses (Dawood et al. 2021; Gaglani et al. 2022; Naleway et al. 2023).

Most HCP participants ($\geq 68\%$) in this study had received egg-based influenza vaccination during the preceding five seasons. The frequent prior repeat vaccinations could lead to higher baseline titers and also blunt antibody responses to vaccination in Year 1 as reflected by the limited fold-rise of MN or HI antibody titers (Figure 3). Regardless, RIV4 outperformed IIV4 and cclIV4 vaccines by inducing significantly

higher fold-rise in antibody titers to A(H3N2) cell vaccine virus, and overall higher fold-rise to A(H1N1)pdm09 and B cell vaccine viruses, providing improved vaccine immunogenicity to mitigate the impact of the prior egg-based vaccines even in a population with background of frequent egg-based influenza vaccinations.

8. The role of egg vs cell-based vaccine in 2 years is interesting. What is the role of antigenic sin and how does it impact the immune response. Please discuss further with current literature and how to overcome with better vaccine approaches going forward?

Response:

Original antigenic sin (OAS) or immune priming can play dual roles and is largely directed by the immunodominance of imprinted HA epitopes upon first exposure to influenza antigen. In this study, the egg-adapted changes on the HA head domain in the egg-grown A(H3N2) vaccine virus: T160K (loss of glycosylation at 158), L194P, and D225G resulted in an “antigenic mismatch” from circulating strains that likely contributed to the low observed VE during the 2016-17 to 2018-19 seasons (Liu et al. 2021, Wu et al, 2017, Zost et al, 2017). It is evident that this set of egg-adaptive changes became immunodominant as we previously reported that upon receipt of the egg-based influenza vaccine, immunologically naïve children developed MN antibodies mostly targeting these egg-adaptive sites with undetectable or minimal MN titers to cell-grown A(H3N2) vaccine virus (Liu et al. JCI 2021). In some adult and older adults, memory B cells targeting the unglycosylated HA 158 epitope was likely first imprinted in their childhood since the A(H3N2) strains circulated between 1968 and 2014 did not have glycosylation at HA 158 site. We have previously reported that egg-based vaccine in 2016-17, 2017-18, and 2018-19 season which contained the same egg-adapted changes on H3 HA head, preferentially boosted MN/HI antibodies targeting these egg-adaptive sites in a broader age range (6 month old to 65+ years) (Liu et al. JCI 2021).

To address this comments, we have added additional discussion points, including how the current finding can provide scientific evidence to develop improved vaccination strategies in the revision:

“Human immunity to influenza is quite complex and present challenges to improve vaccination effectiveness. Immune priming can play dual roles and is largely directed by the immunodominance of imprinted HA epitopes upon first exposure to influenza antigen, as well as the repeated influenza exposures by vaccinations or infections later in lifetime (Liu, 2018 #309;Liu, 2021 #453). Here, our study demonstrated how immunodominant egg-adapted epitopes on A(H3N2) HA can impact the immunogenicity from repeated vaccination. Collectively with previous reports, findings from this study could provide a few meaningful insights to improve vaccination strategy. First, it would be important not to prime naive children with egg-adapted epitopes, it might be necessary immunize naive children with non-egg-based influenza vaccines. Second, for those who were repeat vaccinated with egg-based vaccines such as HCPs and older adults, vaccination with non-egg-based vaccines could be especially beneficial. In addition, improved vaccine formulations, such as higher vaccine doses and the use of adjuvants, could also improve vaccine immunogenicity. “

9. Earlier studies and meta analysis have postulated that repeat vaccination given every year led to reduced antibody response, quality and vaccine efficacy. Can authors discuss if repeat vaccination (egg or

cell) leading to reduced VE is possibly due to enhanced disease. Like what was speculated during the 2009 H1N1 pdm09 in Canada (Skowronski et al) or enhance influenza disease following mismatched H1N1pdm09 virus in animal models (PMID: 23986398).

Response:

Thanks for the comment. Our findings in this study have no implications regarding the possible association between reduced VE and enhanced disease that was reported by Skowronski et al. 2010 in an observational study and Khurana et al. 2013 in the pig model. In our study, we did observe a boost of HA stalk antibodies following vaccination with both egg and non-egg based vaccines. But whether there is any association between the stalk antibody responses and enhanced disease needs to be further evaluated.

We have discussed the impact of repeat vaccination on VE as below:

“Repeat vaccination can lead to reduced VE (Ohmit et al. 2014; Jones-Gray et al. 2023; Ramsay et al 2019). In Year 1, the frequent prior repeat egg-based vaccinations may blunt the antibody responses to the subsequent vaccination as reflected by the limited fold-rise in MN/HI antibody titers (Figure 3). Khurana et al. also observed significantly lower fold-rise to A(H3N2), A(H1N1) and influenza B viruses in those with prior vaccination irrespective of vaccine platforms received. 4 In a highly vaccinated population such as HCPs, an effective vaccine should be able to overcome issues with both repeated vaccination and vaccine egg-adapted mutations. RIV4 vaccination reduced the proportions of participants with MN GMT egg/cell ratio ≥ 4 against A(H3N2) by two-fold in both age groups (Figure 5C and 5D). In Year 2, all four repeat vaccination arms with non-egg-based vaccines (ccIIIV-ccIIIV4, ccIIIV4-RIV4, RIV4-ccIIIV4, RIV4-RIV4), further reduced the proportion of participants with HI GMT egg/cell ratio ≥ 4 against A(H3N2) to less than 10% (Figure 7), of which three in 18-44 yrs group (ccIIIV4-RIV4, RIV4-ccIIIV4, RIV4-RIV4) and two in 45-64 years group (ccIIIV4-RIV4, RIV4-RIV4) induced significantly higher HI antibodies to A(H3N2) cell virus than IIV4-IIV4 arm (Figure 6). Collectively, these findings suggest that multiple seasons of vaccination with non-egg-based vaccines may be needed to overcome the pre-existing immune memory to egg-adapted epitopes and re-direct the neutralizing antibody responses towards epitopes on cell-grown viruses that better represent circulating strains.

Furthermore, the difference in antigenicity and egg-adaption between vaccine viruses from consecutive influenza seasons could further confound the effect of repeat vaccination. The A(H3N2) vaccine virus in Year 2 was updated to a 3C.3a A/Kansas/14/2017-like virus, which is genetically and antigenically distant from the 3C.2a vaccine virus A/Singapore/INFIMH-16-0019/2016 in Year 1. Year 2 vaccine virus A/Kansas/14/2017 virus bears different egg-adapted changes at D190N and N246T but possess 194L and 225D that are the same as cell-propagated A(H3N2) vaccine virus in Year 1 (Table 1). This could have also contributed to the improved antibody response to the cell-grown A(H3N2) vaccine virus after the repeat vaccination in year 2. Thus far, most reported VE from repeated vaccination studies were from egg-based vaccines, evaluation of whether repeat vaccination with non-egg-based vaccines will improve VE, especially for A(H3N2), is warranted.”

10. Authors should also provide perspective on limitations and gaps that can help the move the field forward and clinical relevance of these findings for designing better vaccine approaches to address the

key issue of improving vaccine efficacy against Influenza. Do these findings have any role in development of next generation flu vaccines?

Response:

This is a great comment. We believe this current study and other studies provide direct scientific evidence to develop improve vaccination strategies. We have added additional discussion in the main text as below:

“Multiple immune mechanisms can contribute to the protection from influenza infection. Broadly cross-reactive HA stalk antibodies have been considered as one of the targets for universal vaccine development. Here, we assessed both H1 and H3 HA stalk antibody responses in all vaccine groups in both years. When comparing the post-vs pre-vaccination HA stalk antibody levels, vaccination induced HA stalk antibody rise in several vaccine groups and repeated vaccination arms, however the levels of fold rise were low (close to 1, Figure 4 and Figure 8) with no significant difference in fold rise among different vaccine group in both years, even in RIV4 group that received 3 times more antigen doses. The difference in HA stalk antibody titers post-vaccination mostly reflect the difference in pre-existing stalk antibody levels. To this end, development of next generations vaccines that can induce multiple arms of immune responses could improve the effectiveness of the vaccines.”

“Human immunity to influenza is quite complex and present challenges to improve vaccination effectiveness. Immune priming can play dual roles and is largely directed by the immunodominance of imprinted HA epitopes upon first exposure to influenza antigen, as well as the repeated influenza exposures by vaccinations or infections later in lifetime (Liu, 2018 #309;Liu, 2021 #453). Here, our study demonstrated how immunodominant egg-adapted epitopes on A(H3N2) HA can impact the immunogenicity from repeated vaccination. Collectively with previous reports, findings from this study could provide a few meaningful insights to improve vaccination strategy. First, it would be important not to prime naive children with egg-adapted epitopes, it might be necessary immunize naive children with non-egg-based influenza vaccines. Second, for those who were repeat vaccinated with egg-based vaccines such as HCPs and older adults, vaccination with non-egg-based vaccines could be especially beneficial. In addition, improved vaccine formulations, such as higher vaccine doses and the use of adjuvants, could also improve vaccine immunogenicity. ”

“Taken together, our study demonstrated that multiple seasons of vaccination with non-egg-based influenza vaccines can improve HA-mediated antibody immunogenicity to cell-propagated viruses representing circulating strains. Our findings support the use of antibody responses to cell-propagated viruses, rather than to egg-propagated vaccine viruses, as the primary endpoint in future immunogenicity studies for licensure or comparison of influenza vaccines. Influenza product and platform specific VE estimates could help confirm findings from comparative immunogenicity studies and provide additional information to identify optimal vaccination strategies to prevent influenza.”

Minor

Comment:

1. In results section, authors should add GMT titers for each group when describing the results for easy reading and comparisons.

Response:

Thanks for the careful review. We have added GMT titers accordingly.

Reviewer #2 (Remarks to the Author):

This is a terrific study from Liu and CDC colleagues characterizing human antibody responses following vaccination with egg-based, cell-based, and recombinant protein-based influenza vaccines. Consistent with previous studies, the authors show that the recombinant protein-based vaccines (which possess 'wild type' immunogens) elicit higher neutralizing antibody titers against 'wild type' viruses, especially H3N2. Importantly, the authors measure antibody responses over 2 consecutive seasons with participants receiving different cell/egg/recombinant protein vaccines. They find that vaccination with cell and recombinant protein vaccines can elicit high levels of antibodies against wild type viruses in individuals who previously received egg-based vaccines, and there seems to be an additive effect when cell and recombinant protein vaccines were delivered in consecutive years (in a population previously vaccinated many times with egg-based vaccines). The study is important—the authors nicely describe caveats of the experiments and how it is unknown how the serological data will relate to VE—however even with these caveats, it seems pretty clear to me that we should abandon egg-based influenza vaccines and increase our efforts to produce large amounts of recombinant protein based vaccines. This is a very important study that should shape public policy.

1. The different amount of antigen in each vaccine is discussed at the end, but this information should also be provided in the introduction for readers that might not know this.

Response:

Thanks for pointing this out. We have stated the difference of HA antigen amount in each vaccine in the Introduction on page 3:

“RIV4 contains 3 times the antigen dose at 45ug HA/dose/strain compared to the standard dose IIV4 and cclIV4 (15ug HA/dose/strain).”

2. It seems odd to present the egg/cell ratio data of Year 2 first before presenting Year 2 antibody data. Please consider switching the order of Figure 6 and 7.

Response:

Thanks for the suggestion. We agreed with the reviewer and have switched the order of Figure 6 and 7 and updated the results accordingly in the revised manuscript on page 10 and 11, it now reads:

“We compared the post-vaccination HI GMTs to the cell-grown vaccine viruses between IIV4-IIV4 (Fluzone) and each of the 6 repeat vaccination arms with non-egg-based vaccines (Figure 6). Participants in the younger age group (18-44 years) in the cclIV4-RIV4, RIV4-cclIV4, RIV4-RIV4, and IIV4-RIV4 arms, and the older age group (45-64 years) in cclIV4-RIV4 and RIV4-RIV4 arms, all mounted significantly higher ($p < 0.05$) neutralizing antibody titers to cell-A(H3N2) virus than those in the IIV4-IIV4 (Fluzone) arm. For HI

antibody response to A(H1N1) cell virus, RIV4-RIV4 and IIV4-RIV4 arms in 18-44 years, cclIV4-RIV4, RIV4-cclIV4, RIV4-RIV4 arms in 45-64 years all had significantly higher ($p<0.05$) HI antibody responses than those in the IIV4-IIV4 (Fluzone) arm. Vaccinees in the younger age group (18-44 years) in the cclIV4-RIV4, RIV4-RIV4, and IIV4-RIV4 arms had significantly higher ($p<0.05$) HI antibody responses to both B/VIC and B/YAM cell viruses versus those in the IIV4-IIV4 (Fluzone) arm. Participants in the older age group (45-64 years) who received non-egg-based vaccine in year 2, most had significantly higher ($p<0.05$) HI GMTs to both B/VIC and B/YAM cell virus compared to those in the IIV4-IIV4 (Fluzone) arm, except cclIV4-cclIV4 and IIV4-cclIV4 arms for B/YAM cell virus (Figure 6). It is worth noting that those who received IIV4 in Year 1 then RIV4 in Year 2 demonstrated improved antibody responses to cell-grown vaccine viruses compared to those who received IIV4-IIV4 (Fluzone). The seropositivity rates (Supplementary Table S4) and seroconversion rates (Table S5) to both egg- and cell- viruses in year 2 are summarized in the supplementary materials, and reported previously 2, 3. Overall, in year 2 participants who received one or two non-egg-based vaccines mounted higher post-vaccination HI antibody responses to cell-grown vaccine viruses than those who received repeated egg-based IIV4 vaccination in both years (IIV4-IIV4).

At 1-month post-vaccination during Year 1, HCPs who received only one season of non-egg-based vaccine including RIV4 still had 21-40% with A(H3N2) GMT egg/cell titer ratio ≥ 4 (Figure 5). We therefore investigated whether repeat vaccination with non-egg-based vaccines in year 2 could further reduce the HI GMT egg/cell titer ratio (Figure 7). Among recipients of IIV4-IIV4 (Fluzone), the proportion of participants with HI GMT egg/cell titer ratio >4 generally increased, especially for A(H3N2) in the 18-44 years group, B/VIC virus and B/YAM virus in 45-64 years group, suggesting repeated boost to antibodies targeting egg-adapted epitopes in these participants after receiving consecutive egg-based IIV4 vaccination. In contrast, among recipients of non-egg-based vaccines in just one study year (IIV4-cclIV4 or IIV4-RIV4) or in both study years (cclIV4-cclIV4, cclIV4-RIV4, RIV4-cclIV4, RIV4-RIV4), the proportions of participants with egg/cell ratio ≥ 4 to generally decreased post-vaccination though did not reach statistical significance in some repeat vaccination groups (Figure 7). Among those, the cclIV4-RIV4 arm had significant ($p<0.05$) reduction in the proportion of HI GMT egg/cell ratio ≥ 4 against A(H3N2) viruses comparing pre- vs post-vaccination in the 45-64 years group, and the RIV4-cclIV4 arm had significant ($p<0.05$) reduction in the proportion of GMT egg/cell ratio ≥ 4 against A(H1N1)pdm09 virus in the younger age group in year 2, suggesting multiple seasons of repeat vaccination with non-egg-based vaccines may be needed to overcome the dominant antibody responses to the egg-adapted epitopes and redirect the antibody responses away from the egg-adapted epitopes.”

3. Were experiments completed to measure antibody titers to Year 1 vaccine strains with Year 2 serum samples? It might be interesting to see if the antigenically distinct Year 2 strains boosted antibodies to the Year 1 viruses. Consider adding these data if these experiments have already been completed.

Response:

Thanks for the great comment. We did not test Year 2 serum samples against the Year 1 vaccine strains in the current scope of this study. In this study, B vaccine viruses remained the same for both years.

While A(H1N1)pdm09 vaccine strains were updated in Year 2, they were antigenically very similar to A(H1N1)pdm09 vaccine strains used in Year 1. A(H3N2) vaccine strains were the only component that was antigenically distinct between Year 1 and Year 2.

However, although beyond the scope of the current study, for each influenza season, our group at CDC analyzes vaccine sera against panels of both historic and emerging drifted influenza viruses. We do often see “back boost” of the old vaccine strain (e.g. from year 1) when vaccinated with antigenically distinct updated vaccine strains (e.g. in year 2), while the antibody responses to the current vaccine viruses (e.g. year 2) are not compromised.

4. Line 177 on pg 8 should be revised because this doesn't appear to be the case with H3 responses in 45-64 year olds.

Response:

Thanks for the careful review. We have revised the statement accordingly which is cited below:

“especially for A(H3N2) in 18-44 years group, B/VIC virus, and B/YAM virus in 45-64 years group,”

Reviewers' Comments:

Reviewer #1:

Remarks to the Author:

The authors have diligently addressed this reviewers' comments. The additional data improves the manuscript significantly.

Congrats on a nice study.